# Single neurons and networks in the mouse claustrum integrate input from widespread cortical sources

Andrew M Shelton[1]*, David K Oliver[1], Ivan P Lazarte[1], Joachim S Grimstvedt[2], Ishaan Kapoor[1], Jake A Swann[1], Caitlin A Ashcroft[1], Simon N Williams[1], Niall Conway[1], Selma Tir[3], Amy Robinson[2], Stuart Peirson[3], Thomas Akam[4], Clifford G Kentros[2], Menno P Witter[2], Simon JB Butt[1]*, Adam Max Packer[1]*

[1]Department of Physiology, Anatomy, and Genetics, University of Oxford, Oxford, United Kingdom; [2]Kavli Institute for Systems Neuroscience, Centre for Neural Computation, Norwegian University of Science and Technology, Trondheim, Canada; [3]Nuffield Department of Clinical Neurosciences, Sir Jules Thorn Sleep and Circadian Neuroscience Institute (SCNi), University of Oxford, Oxford, United Kingdom; [4]Department of Experimental Psychology, University of Oxford, Oxford, United Kingdom

*For correspondence:
andrew.shelton1312@gmail.com (AMS);
simon.butt@dpag.ox.ac.uk (SJBB);
adampacker@gmail.com (AMaxP)

**Competing interest:** The authors declare that no competing interests exist.

## eLife Assessment

This study compiles a wide range of results on the connectivity, stimulus selectivity, and potential role of the claustrum in sensory behavior. While most of the connectivity results confirm earlier studies, this **valuable** work provides **incomplete** evidence that the claustrum responds to multi-modal stimuli and that local connectivity is reduced across cells that have similar long-range connectivity. The conclusions drawn from the behavioral results are weakened by the animals' poor performance on the designed task. This study has the potential to be of interest to neuroscientists.

**Abstract** The claustrum is thought to be one of the most highly interconnected forebrain structures, but its organizing principles have yet to be fully explored at the level of single neurons. Here, we investigated the identity, connectivity, and activity of identified claustrum neurons in *Mus musculus* to understand how the structure's unique convergence of input and divergence of output support binding information streams. We found that neurons in the claustrum communicate with each other across efferent projection-defined modules which were differentially innervated by sensory and frontal cortical areas. Individual claustrum neurons were responsive to inputs from more than one cortical region in a cell-type and projection-specific manner, particularly between areas of frontal cortex. In vivo imaging of claustrum axons revealed responses to both unimodal and multimodal sensory stimuli. Finally, chronic claustrum silencing specifically reduced animals' sensitivity to multimodal stimuli. These findings support the view that the claustrum is a fundamentally integrative structure, consolidating information from around the cortex and redistributing it following local computations.

## Introduction

The claustrum (CLA) - defined by its dense connectivity with the cerebral cortex - has been implicated in a variety of the sensory and cognitive functions, including sleep (*Narikiyo et al., 2020*; *Norimoto*

*et al., 2020*; *Marriott et al., 2024*), saliency detection (*Mathur, 2014*; *Remedios et al., 2014*; *Smith et al., 2019*), multisensory integration (*Crick and Koch, 2005*; *Smythies et al., 2012*; *Smythies et al., 2014*; *Vidyasagar and Levichkina, 2019*), and task engagement (*Atlan et al., 2021*; *Fodoulian et al., 2020*; *Jackson et al., 2018*; *Dillingham et al., 2017*; *Goll et al., 2015*; *Faig et al., 2024*). Lesion and anatomical studies point to a multifunctional role (*Atilgan et al., 2022*; *Edelstein and Denaro, 2004*; *Patru and Reser, 2015*), with CLA reciprocally connected with large swaths of cortex in a projection pattern that has been previously described as a 'crown of thorns' (*Atlan et al., 2017*; *Peng et al., 2020*; *Wang et al., 2023*; *Zingg et al., 2014*). Recent research has made it increasingly clear that this connectivity is not uniform, with CLA preferentially projecting to prefrontal and midline cortical regions (*Wang et al., 2023*; *Zingg et al., 2014*; *Zingg et al., 2018*). These projections appear to be organized into connectivity-defined modules within the CLA (*Chia et al., 2020*; *Marriott et al., 2021*; *Gattass et al., 2014*; *Remedios et al., 2010*), leading to functional selectivity (*Atlan et al., 2021*; *Chevée et al., 2022*). However, it has yet to be determined the extent to which this connectivity is specified and combined at the level of single CLA neurons. Here, we seek to address this deficit by providing a detailed analysis of CLA circuits and cell types from intraclaustral, corticoclaustral, and claustrocortical perspectives, thereby building a comprehensive platform for our future understanding of CLA function.

Despite the difficulties presented by the challenging anatomy and broad connectivity of the CLA (*Zingg et al., 2018*), progress in identifying CLA cell types and circuit motifs in vitro has illuminated some aspects of how CLA neurons are wired and the implications this may have for their computations (*Chia et al., 2020*; *Graf et al., 2020*; *Kim et al., 2016*; *Qadir et al., 2022*). For example, cortical projections to the CLA target efferent-defined modules and activate a dense network of feedforward inhibition via local interneurons (*Chia et al., 2020*; *Kim et al., 2016*; *Ham and Augustine, 2022*; *LeVay and Sherk, 1981*; *Smith et al., 2012*; *Smith and Alloway, 2010*). Recently, it has been discovered that cell types in the CLA of mice are responsive to inputs from more than one cortical area (*Qadir et al., 2022*; *Yuan et al., 2015*). However, these findings specifically address the input of only one cortical area onto single CLA neurons at a time and do not address which CLA neurons may be summing inputs from different areas in the same experiment. Nor do they address whether any computations within the CLA itself can occur, given the reach and strength of local inhibitory networks and the apparent sparsity of excitatory-excitatory connectivity in the coronal plane (*Kim et al., 2016*; *Orman, 2015*). Although several studies have thoroughly documented the dense connectivity of the CLA (*Chia et al., 2020*; *Qadir et al., 2022*; *Ham and Augustine, 2022*), the question of whether individual CLA neurons participate in a single, dedicated network or in potentially several networks remains unresolved and has broad implications for our further understanding of CLA function.

Here, we employed a retrograde labeling strategy to unequivocally identify a specific subpopulation of retrosplenial-projecting CLA neurons (CLA$_{RSP}$) and characterize their intrinsic electrophysiological and morphological properties among other CLA neurons. We then leveraged our knowledge of this population to understand how projection- and electrophysiologically defined CLA cell types map onto CLA connectivity by investigating intraclaustral, corticoclaustral, and claustrocortical circuits. Finally, we measure the activity of CLA$_{RSP}$ neurons in vivo and investigate how chronic CLA silencing affects behavior in a variety of assays. The results obtained through this investigation provide evidence of the CLA as an intrinsically multimodal structure, with single neurons being capable of combining and therefore integrating afferent and local input in a manner that is dependent on cell type and projection target.

## Results

### Anatomical delineation of CLA in mice via retrograde tracing

We first sought to define a specific subset of CLA neurons based on their efferent connectivity. To label CLA neurons in the adult mouse, we used a retrograde tracing strategy by injecting fluorescently labeled cholera toxin subunit B (CTB) into the retrosplenial cortex (RSP), which receives input specifically from the CLA and no CLA-adjacent structures (*Zingg et al., 2018*; *Marriott et al., 2021*). To understand how CLA neurons differentially innervate RSP, we injected CTB488 (green), CTB555 (red), CTB647 (blue) into three separate rostrocaudal locations along the RSP (n=3 mice, *Figure 1A*). Confocal microscopy imaging of 100 μm-thick coronal sections revealed that each injection labeled

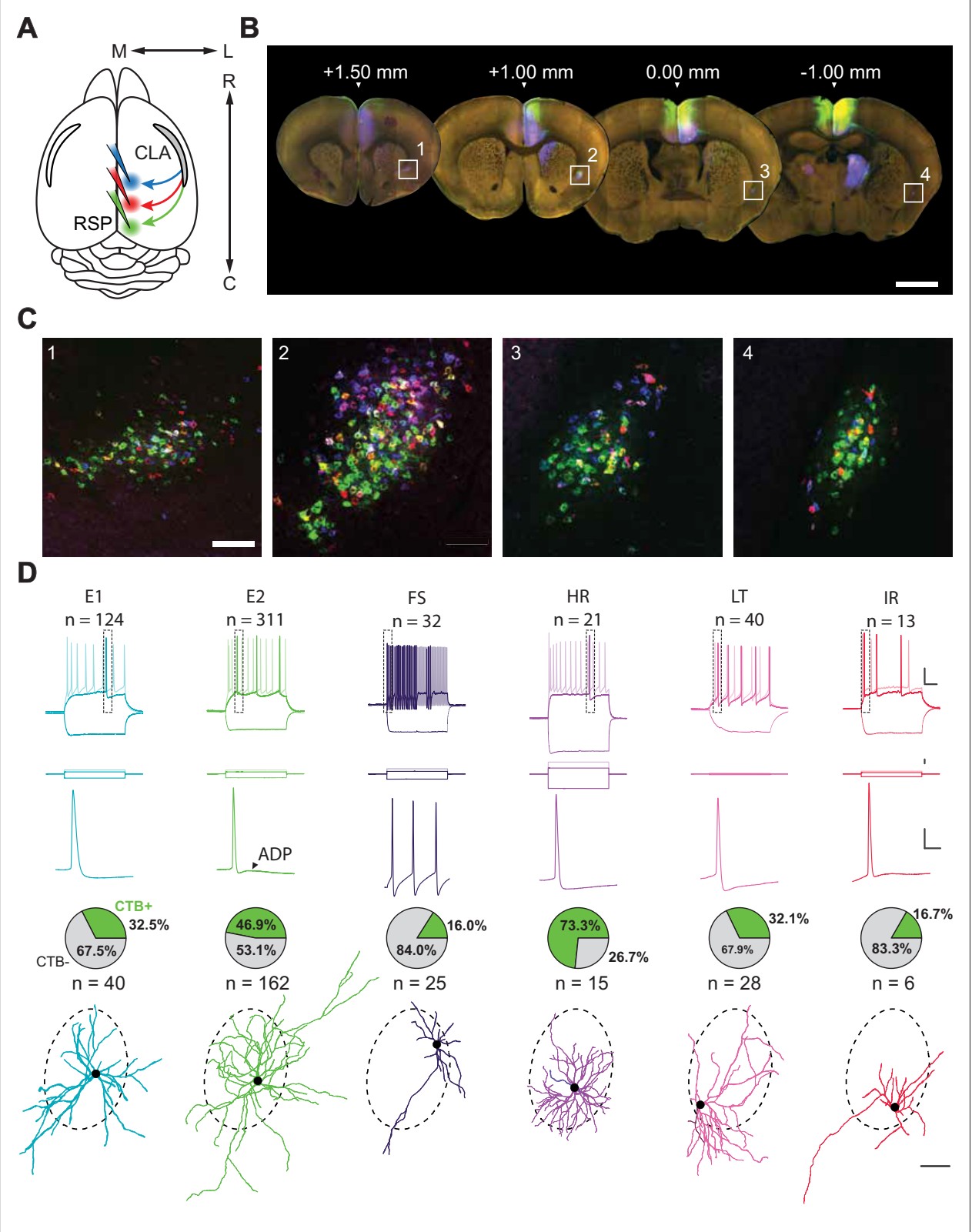

**Figure 1.** Interrogation of a projection-defined CLA neuronal subpopulation. (**A**) Schematic of the injection strategy. Three different CTB-AlexFluor conjugates were injected into separate rostrocaudal regions of RSP. (**B**) Representative 100 µm sections spanning the rostrocaudal axis of the brain in which the CLA can be seen retro-labeled by CTB from rostral (blue), intermediate (red), and caudal (green) injections into RSP (n=3 mice). Scale bar = 1 mm. (**C**) Insets of **B** in which CLA neurons are labeled with CTB. Scale bar = 200 µm. (**D**, top to bottom) Intrinsic electrophysiological profiles,

*Figure 1 continued on next page*

*Figure 1 continued*

expanded spike waveforms (from top inset), proportion of neurons found to be CTB +during experiments (cells for which CTB status was known), and example morphological reconstructions for each electrophysiological cell type. Scale bars (top to bottom): 20 mV/100ms, 300 pA, 20 mV/10ms, 100 μm. Dashed border around morphologies represents the average CLA$_{RSP}$ area across slices from *Figure 1—figure supplement 2*. Somas aligned to their approximate location in this region during patching.

The online version of this article includes the following figure supplement(s) for figure 1:

**Figure supplement 1.** Identification of the CLA.

**Figure supplement 2.** Confocal image processing and analysis.

**Figure supplement 3.** Methods of single-cell electrophysiology.

**Figure supplement 4.** Unsupervised identification of electrophysiological cell types in the CLA.

**Figure supplement 5.** Mapping.

**Figure supplement 6.** All CTB +interneurons encountered during in vitro whole-cell patch-clamp experiments, arranged by type.

**Figure supplement 7.** Quantification of retrograde label expression in the CLA.

spatially overlapping populations of CLA neurons, distinct from the surrounding, unlabeled tissue (*Figure 1B, C*, *Figure 1—figure supplement 1*). A comparison between injection sites of CTB labeling in CLA across mice showed that the caudal injection site reliably labeled the most cells overall, especially in caudal regions of CLA (*Figure 1—figure supplement 1*). Further, this strategy consistently demonstrated that CLA$_{RSP}$ neurons could be found along the whole CLA length, as defined by other sources (*Franklin and Paxinos, 2008*; *Wang et al., 2017*; *Wang et al., 2020*; *Grimstvedt et al., 2023*), irrespective of RSP injection location. Moreover, the level of overlap between CLA neurons projecting to different regions of RSP varied depending on the injection site (*Figure 1—figure supplement 1*). We chose the caudal-most RSP injection site for the remainder of the study due to the dense and highly specific labeling of CLA$_{RSP}$ neurons.

We next compared known markers of the CLA against the labeling of CLA$_{RSP}$ neurons using immunohistochemistry to obtain a better understanding of the intraclaustral localization of CLA$_{RSP}$ neurons *Wang et al., 2023*; *Grimstvedt et al., 2023*; *Druga et al., 1993*; *Real et al., 2003*; *Real et al., 2006*; *Shaker et al., 2024* (n=3 mice, *Figure 1—figure supplement 1*, for further analyses, see also *Grimstvedt et al., 2023*). CLA$_{RSP}$ neurons were aligned with the parvalbumin (PV) neuropil-rich CLA 'core' and a paucity of myelinated axons in the same region, as shown by myelin basic protein (MBP). MBP also labeled densities of myelinated axons above and below the PV plexus, indicating dorsal and ventral aspects of the CLA. Therefore, CLA$_{RSP}$ neurons were taken to represent an efferent-defined subpopulation of CLA neurons.

Analysis of thin (50 μm), sequential coronal sections demonstrated that the density of CLA$_{RSP}$ neurons varied along the rostrocaudal axis (n=3 mice, *Figure 1—figure supplement 2*) with the highest number of CLA$_{RSP}$ neurons found around 1 mm rostral to bregma (1070±261 cells/animal, 27±11 cells/section; *Figure 1—figure supplement 2*). The rostral and caudal poles of CLA differed in that rostrally-located CLA$_{RSP}$ neurons were found at low density, that is showing a dispersed distribution, whereas neurons found 1 mm caudal to bregma were densely packed in a relatively small cross-section of tissue. Together, these anatomical experiments enabled us to define and target a specific group of CLA$_{RSP}$ neurons in subsequent electrophysiological investigations.

## Intrinsic electrophysiological characterization of CLA neurons reveals distinct subpopulations

To extend and better understand the specificity of the putative CLA$_{RSP}$ module, we repeated our retrograde labeling strategy, but this time in conjunction with acute in vitro whole-cell patch-clamp electrophysiology to explore heterogeneity in CLA$_{RSP}$ neurons and non-CLA$_{RSP}$ within the claustrum (*Figure 1D*, *Figure 1—figure supplements 3–4*, see Materials and methods). Recovered morphologies were matched with intrinsic electrophysiological profiles using a standardized, quality-controlled protocol for a final dataset of 540 neurons (*Figure 1—figure supplement 3*). We identified several subtypes of both putative excitatory (total number of cells recovered: n=434) and inhibitory (n=106) neurons based on intrinsic electrophysiological properties (*Figure 1D*).

Delineation of some subtypes was validated by unsupervised clustering on a dimensionally reduced dataset (*Figure 1—figure supplement 4*, *Supplementary file 1*). While a significant proportion of putative excitatory neurons were CLA$_{RSP}$, population-level homogeneity among this group impeded further clustering using unsupervised means. Rather, we relied on several intrinsic electrophysiological features – evident within the action potential waveform and firing pattern for each cell – to define two excitatory cell subtypes (E1 and E2). E1 and E2 neurons could be divided by spike amplitude adaptation normalized from the first action potential: E1 monophasically declined while E2 showed a biphasic pattern, initially declining sharply before recovering slightly (*Figure 1—figure supplement 4*). E2 neurons could be further differentiated from E1 neurons by the presence of an afterdepolarization potential that led to a bursting spike doublet at higher current injections (ADP, 1.9±2.3 mV; *Figure 1—figure supplement 4*, *Supplementary file 1*).

Interneurons, by contrast, were readily categorized into four groups (high rheobase [HR], fast-spiking [FS], low threshold [LT], irregular [IR]) using unsupervised methods alone (average inhibitory silhouette score = 0.853, k=4 clusters; *Figure 1—figure supplement 4*). FS cells were found to fire short half-width duration spikes at frequencies tested up to 200 Hz, while LT cells had a low rheobase and high input resistance. Conversely, HR cells had a large rheobase and low input resistance with a significant delay to spike at threshold. Morphologically, HR interneurons were sparsely spiny (*Figure 1—figure supplement 5*) with a dense dendritic arbor similar to that reported for neurogliaform cells. Finally, IR cells fired irregularly and very infrequently compared to other types. Electrophysiological feature comparisons between these cells additionally supported distinct subtypes that differed from excitatory cells (*Figure 1—figure supplement 4*; *Butt et al., 2005*; *Kawaguchi and Kubota, 1997*). Surprisingly, a small subset of putative interneurons was found to be CTB+ (some of which were co-labeled with tdTomato in Nkx2.1-Cre;Ai9 +animals; *Figure 1—figure supplement 6*), suggesting the presence of inhibitory projection neurons within the CLA. These cells, the majority of which were HR neurons, represented 23% of our putative inhibitory subtypes and 5% of total CLA neurons (*Figure 1D*, *Figure 1—figure supplement 6*). Further to this, we used immunohistochemistry to independently confirm that GABAergic cells were captured using retrograde labeling approaches (*Figure 1—figure supplement 7*).

We reconstructed 134 recovered morphologies (*Figure 1—figure supplement 5*) and found the majority of E1 and E2 subtypes had spiny dendrites, consistent with them being excitatory neurons (*Figure 1—figure supplement 5*). FS, HR, LT, and IR types were either aspiny or sparsely spiny in line with cortical GABAergic interneurons. This distinction aside, classical morphological analyses alone did not adequately define CLA cell types, again highlighting the need for connectivity- and function-defined approaches (*Figure 1—figure supplement 5*, *Supplementary file 2*). Overall, our patch-clamp recorded neurons expand on previous knowledge of CLA neuron diversity *Graf et al., 2020*; *Qadir et al., 2022*, revealing a mix of excitatory and inhibitory neurons within this nucleus. No neuronal subtype was found to be exclusive to the CLA$_{RSP}$ module, suggesting that efferent connectivity is not subtype specific.

## Intraclaustral projections favor a cross-modular arrangement

Given that CLA$_{RSP}$ versus non-CLA$_{RSP}$ neurons are indistinguishable based on intrinsic parameters, we next explored if excitatory synaptic connections exist amongst these neurons in the CLA$_{RSP}$-defined region, as previous studies - employing a variety of approaches - have failed to reach consensus on this issue *Zingg et al., 2018*; *Kim et al., 2016*; *Orman, 2015*; *Smith and Alloway, 2014*. We used a dual retrograde Cre (retroAAV-Cre) and CTB injection strategy in the RSP combined with conditional viral expression of AAV-FLEX-ChrimsonR-tdTomato in CLA (*Figure 2A*). We then photostimulated ChrimsonR +presynaptic axon terminals throughout the rostrocaudal length of CLA while recording from either CTB + or CTB- CLA neurons (*Figure 2B and C*) and restricted analysis to monosynaptic connections with latencies of 3–12ms to remove CLA neurons directly expressing opsin from the dataset (*Figure 2—figure supplement 1*, see Materials and methods), as reported by studies of opsin kinetics *Li et al., 2017*; *Melzer et al., 2012*; *Petreanu et al., 2007*. We found that we could evoke short-latency, putative monosynaptic excitatory postsynaptic potentials (EPSPs) in the majority (69.5%; n=32/46) of recorded CLA neurons, although only a small subset of these were CLA$_{RSP}$ neurons (n=4/11 CTB + cells responsive; *Figure 2D*, *Figure 2—figure supplement 1*). In addition, we found that both excitatory and inhibitory neuronal subtypes exhibited EPSPs in response to CLA$_{RSP}$

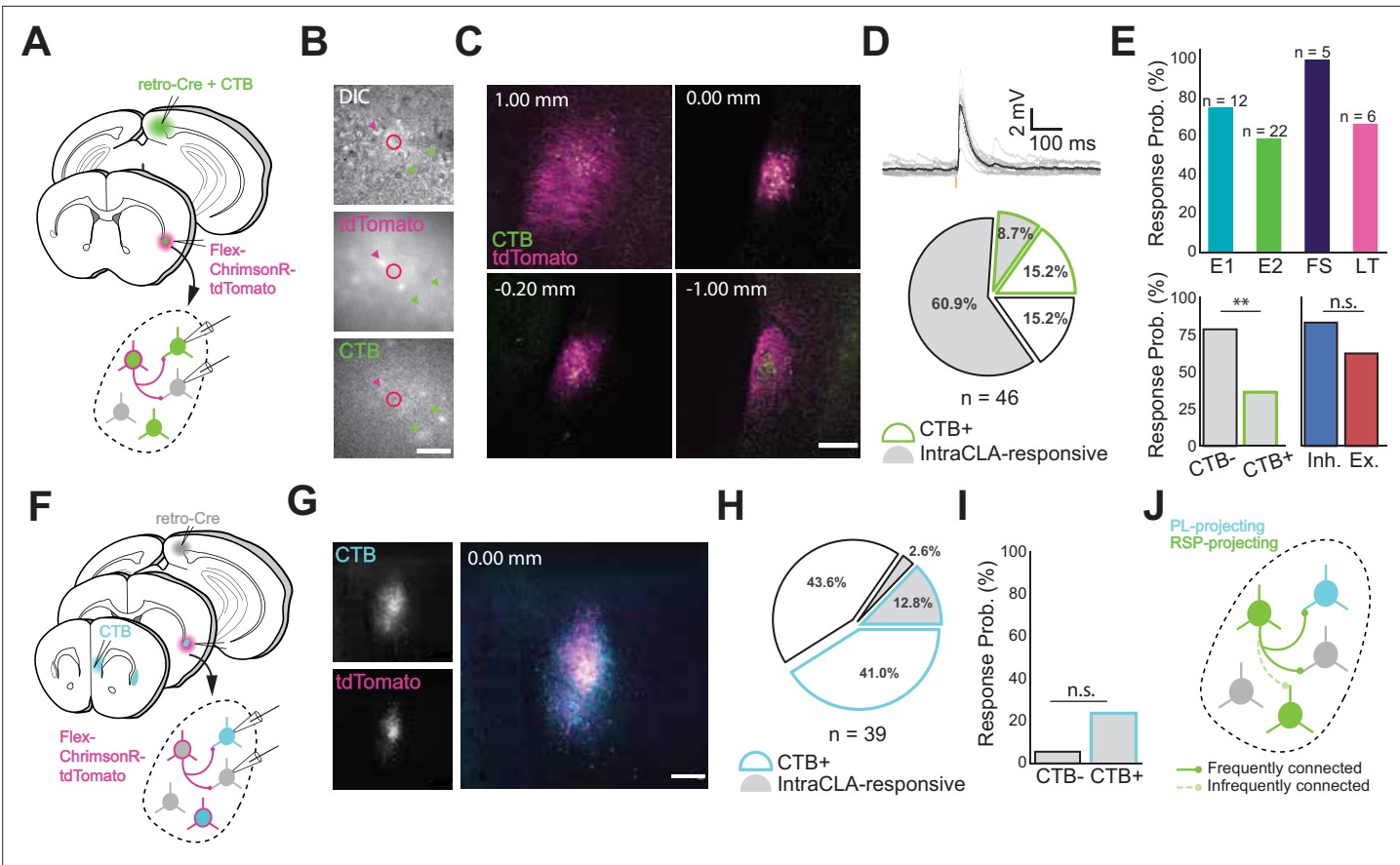

**Figure 2.** Intraclaustral connectivity is common and cross-modular. (**A**) Schematic of injection and patching strategy in the CLA. (**B**) Transmitted light images taken during patching. Magenta arrows indicate ChrimsonR-tdTomato-expressing neurons while green arrows indicate CTB-labeled neurons. Note that these populations only partially overlap. Scale bar = 50 μm. (**C**) Expression of AAV-FLEX-ChrimsonR-tdTomato and CTB in the CLA along the rostrocaudal axis. Notice the lack of tdTomato + cell bodies in the far rostral and far caudal sections beyond the spread of the virus. Scale bar = 200 μm. (**D**) CLA neurons frequently displayed EPSPs in response to presynaptic CLA neuron photostimulation with 595 nm light (top). CLA neurons not labeled with CTB were found to be proportionally the most likely to respond to presynaptic photostimulation (n=28/46 neurons, bottom). (**E**, top) Most electrophysiological types were represented among neurons responsive to CLA input (n is the number of neurons recorded in each group, total n=32 responsive/46 neurons). No significant difference in response probability was found between excitatory and inhibitory cell types (p=0.29, Fisher exact test), while a significant difference in responsivity was found between CTB + and CTB- neurons (p=0.009, Fisher Exact Test) (bottom). (**F**) Schematic of injection and patching strategy in the CLA. (**G**) Confocal image of CTB and ChrimsonR-tdTomato expression in the CLA from separate injections into PL and RSP. Scale bar = 200 μm. (**H**) Quantification of responses from experiments in (**F**). CLA neurons that were labeled with CTB from PL were more likely to respond to CLA$_{RSP}$ input than those that weren't, but not by a significant margin (**I**, p=0.19, Fisher Exact Test). (**J**) Model of the circuit investigated, showing that CLA$_{RSP}$ neurons are more likely to form active synapses with non-CLA$_{RSP}$ neurons.

The online version of this article includes the following figure supplement(s) for figure 2:

**Figure supplement 1.** Measurement of monosynaptic responses in the CLA.

**Figure supplement 2.** Voltage imaging of CLA activity propagation.

optogenetic stimulation (*Figure 2E*, bottom right), although there were some differences between subtypes. Of the excitatory cell subtypes, E1 (75.0%) was more likely to receive local CLA$_{RSP}$ input than E2 neurons (59.1%), despite the latter group being the predominant subtype in the CLA (*Figure 2E*, top). Among inhibitory subtypes, FS neurons (100%) exhibited the highest probability of receiving CLA$_{RSP}$ input, while LT neurons (66.7%) received CLA$_{RSP}$ input with probabilities comparable to that of excitatory types. Despite variability in these response probabilities, we found no statistically significant differences between cell types in the likelihood of responding to intraclaustral input (*P*>0.05, Fisher Exact test between all types, Bonferroni corrected). These data suggest – with implications for information transfer within the CLA – that the primary factor underpinning the organization of

intraclaustral connectivity is projection target, that is CLA neurons that project to areas outside of RSP are more likely to receive local input from those that do (CLA$_{RSP}$).

To further examine this, we turned to a different CLA circuit (*Figure 2F*) that involved both prelimbic-projecting CLA neurons (CLA$_{PL}$) and CLA$_{RSP}$. Qualitatively, CLA$_{PL}$ neurons occupied a larger area of the CLA and made up a larger share of CLA neurons overall compared to CLA$_{RSP}$ (*Figure 2G and H*). While fewer CLA$_{PL}$ neurons were CLA$_{RSP}$-responsive than CLA$_{RSP}$-unresponsive (presumably because some of these were also CLA$_{RSP}$ but not expressing opsin), they were not significantly more likely to be CLA$_{RSP}$-responsive than non-CLA$_{PL}$ neurons (*Figure 2I*). Taken together, these experiments point toward a complex intraclaustral circuitry that is predisposed toward inter-module connectivity, for example that CLA neurons receiving input from a given cortical area preferentially target CLA neurons projecting to a different area (*Figure 2J*). How this circuit logic influences intraclaustral computations has critical implications for the signals the CLA transmits downstream.

To address whether such connectivity exists in the rostrocaudal plane, we next turned to a horizontal slice preparation using fluorescent voltage-sensitive dye (VSD RH-795; *Figure 2—figure supplement 2*, n=13 animals, 28 recordings) to examine potential l connectivity in vitro. Electrical stimulation of the rostral pole of the CLA resulted in a traveling wave of increased voltage that spread to caudal CLA over a period of 10–15ms. Blocking glutamate receptors by bath application of DNQX and APV abolished these responses, supporting a role for glutamatergic transmission within and along the rodent CLA. This transmission was bidirectional as electrical stimulation of caudal CLA elicited a wave of depolarization toward the rostral pole with similar temporal properties. These experiments collectively point to extensive and bidirectional intraclaustral connectivity, engaging both excitatory and inhibitory neurons in a manner defined more by efferent target than electrophysiological type per se. This further supports the idea that CLA contains the necessary circuitry to join extraclaustral inputs with a complex and cross-modular internal CLA network.

## Corticoclaustral inputs define a modular spatial organization

Multiple lines of evidence indicate that the CLA contains topographic zones of input and output in mice, yet it remains uncertain how these zones are organized *Atlan et al., 2017*; *Wang et al., 2017*; *Wang et al., 2022*. To resolve this, we again used a retrograde labeling method to distinguish CLA$_{RSP}$, now in tandem with the anterograde viral expression of tdTomato (AAV-ChrismonR-tdTomato) in one of several afferent neocortical areas: frontal (orbitofrontal [ORB], prelimbic [PL], anterior cingulate–anterior part [ACAa], and posterior part [ACAp]), primary motor (MOp), sensory (anteromedial visual cortex [VISam], dorsal auditory area [AUDd]) and parahippocampal (ENTl) cortices (n=3 mice/injection site, n=24 mice in total; *Figure 3A*, *Figure 3—figure supplement 1* see *Table 1*). Coronal sections revealed variation in neocortical axon innervation relative to the CLA$_{RSP}$ region along the dorsal-ventral axis, which was reflected in the likelihood of observing post-synaptic responses to cortical input along this axis (*Figure 2—figure supplement 1*). Several cortical areas, including ACAa and ORB, projected axons medially and laterally to CLA$_{RSP}$ in addition to dorsally or ventrally (*Figure 3C and D*, *Figure 3—figure supplements 1–2*) akin to previously reported domains seen in output neurons of CLA (*Marriott et al., 2021*; *Figure 3E and F*). The distinct dorsoventral patterns of innervation are best exemplified by the complementary projections from ORB, PL, and MOp, which target the ventral, central, and dorsal CLA, respectively (*Figure 3E and F*).

We then investigated the physiological significance of this innervation by optogenetically stimulating presynaptic cortical axon terminals while recording from post-synaptic CLA neurons in vitro (*Figure 3G and H*). We observed short-latency EPSPs in CLA neurons in response to optogenetic stimulation of axons arising from every neocortical injection site (n=266 cells, 93 animals). However, there was variation in the percentage of responsive CLA neurons with stimulation of axons from frontal cortical areas – PL and ACAa having the highest probability of evoking a response in both CLA$_{RSP}$ and non-CLA$_{RSP}$ neurons. Stimulation of axons arising from sensorimotor areas such as AUDd and MOp had the lowest probability of evoking an EPSP with the notable exception of VISam. Further, CLA neurons were more likely to receive input from frontal cortical regions if they projected onward to RSP, that is were CTB+. This relationship was weaker or absent in areas, such as MOp and AUDd, suggesting differences in the input-output routes of these CLA neurons. Results from these experiments confirm the modularity of CLA inputs and how those inputs map onto its outputs but also raise questions regarding how the inputs may be combined onto single CLA neurons.

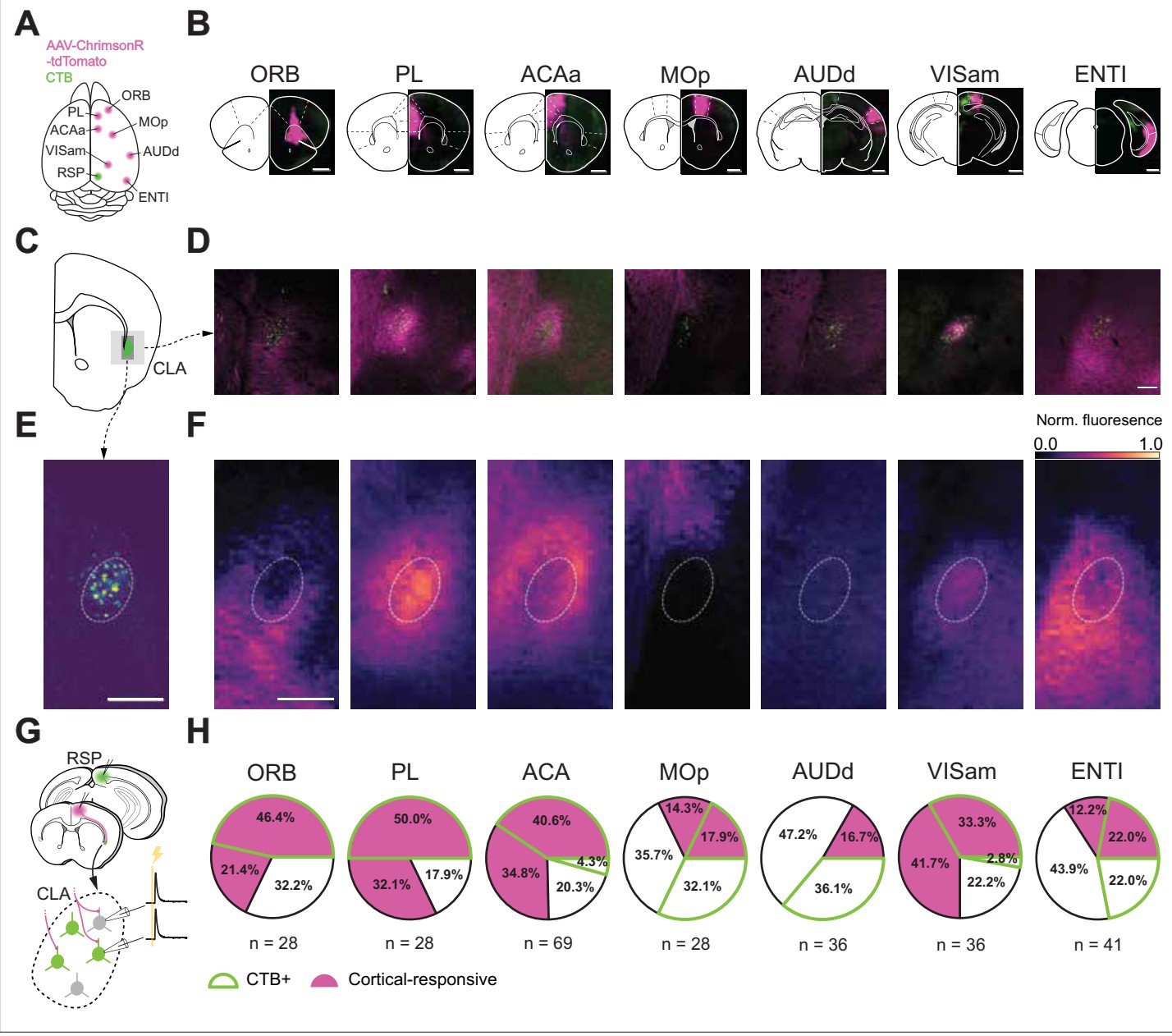

**Figure 3.** Spatial distribution of afferent projections onto the CLA. (**A**) Schematic of injection sites in the cortex. Individual Chrimson-tdTom (magenta) injection sites were combined with an injection of CTB (green) into the RSP. (**B**) Coronal sections of example injection sites. Scale bars = 1 mm. (**C**) Schematics of a representative section of the CLA at 1 mm rostral to bregma. (**D**) Histological sections from the representative section in **C** and each input cortical area in **B**. Scale bar = 200 μm. (**E**) Example image of CLA$_{RSP}$ neurons overlaid with the average contour representing the CLA$_{RSP}$ core from all mice (n=21 mice). (**F**) Heatmaps of normalized fluorescence of corticoclaustral axons in the CLA. Scale bar = 200 μm, 15 μm/pixel. (**G**) Schematic of the patching strategy used to investigate single-cortex innervation of the CLA. (**H**) Individual response and CTB proportions of CLA neurons to each cortex investigated in these experiments. CLA$_{RSP}$ neurons were found to be the most responsive to frontal-cortical input and less responsive to inputs from other regions, with the exception of VISam.

The online version of this article includes the following figure supplement(s) for figure 3:

**Figure supplement 1.** Histology from posterior ACA (ACAp) injections relative to anterior ACA (ACAa; n=3 mice per injection site).

**Figure supplement 2.** Distribution of input axons in the CLA complex.

**Table 1.** Stereotaxic targets and coordinates.
DV coordinates measured as depth from pia.

| Target Area | Abbreviation | AP (mm) | ML (mm) | DV (mm) |
|---|---|---|---|---|
| CLA | CLA | 1.0 | 3.4 | –2.7 |
| Anterior Retrosplenial Cortex | RSPr | –1.50 | 0.50 | –0.75 |
| Intermediate Retrosplenial Cortex | RSPi | –2.25 | 0.50 | –0.75 |
| Posterior Retrosplenial Cortex | RSPc | –3.00 | 0.50 | –1.00 |
| Anterior Anterior Cingulate Cortex | ACAa | 1.34 | 0.30 | –1.25 |
| Posterior Anterior Cingulate Cortex | ACAp | –0.11 | 0.25 | 0.90 |
| Medial Prefrontal Cortex | PL | 1.50 | 0.60 | –1.80 |
| Orbitofrontal Cortex | ORB | 2.50 | 1.20 | –1.80 |
| Primary Motor Cortex | MOp | 0.60 | 1.50 | –0.75 |
| Dorsolateral Entorhinal Cortex | ENTl | –4.30 | 3.50 | –2.25 |
| Secondary Visual Cortex | VISam | –2.70 | 1.50 | –0.50 |
| Dorsal Secondary Auditory Cortex | AUDd | –2.12 | 3.75 | –0.50 |

## Dual-color optogenetic mapping reveals integration of cortical inputs

One of the posited functions of CLA is to affect sensorimotor 'binding' or information integration *Crick and Koch, 2005*; *Edelstein and Denaro, 2004*, defined here as single-cell responsiveness to more than one input pathway, for example being capable of combining and therefore integrating these inputs. Given the distinct topography of input axons (*Figure 3*) and spatial organization of projection targets *Marriott et al., 2021* of the CLA, we next set out to test if single CLA neurons in mice are responsive to more than one cortical region and, therefore, may support established models of CLA function. We combined retrograde tracer injections into the RSP with a dual-color optogenetic strategy, injecting AAV-Chronos-GFP and AAV-ChrimsonR-tdTomato into combinations of the neocortical regions (*Figure 4A*) previously characterized (*Figure 3*). Opsin-fluorophore expression was evident in axons localized in and around the region of CLA$_{RSP}$ neurons during in vitro whole-cell patch-clamp recordings and *post hoc* histology (*Figure 4B*).

Drawing from previously reported methodology *Bauer et al., 2021*; *Hooks et al., 2015* of dual-color optogenetic stimulation, we used prolonged orange light (595 nm, 500ms) to desensitize ChrimsonR opsins and reveal independent blue light-sensitive (470 nm, 4ms) Chronos-expressing input (*Figure 4C*, *Figure 4—figure supplement 1*). Control experiments using one opsin confirmed the viability of this approach (n=6 mice, 21 cells; *Figure 3—figure supplement 1A–C*). Although simultaneous optogenetic stimulation using blue and orange light was possible in these experiments, we did not analyze these data due to the photosensitivity of ChrimsonR to blue light potentially confounding interpretations of EPSP magnitude *Klapoetke et al., 2014*. We found that a subset of CLA neurons receives inputs from more than one cortical area (66/259 of all tested cells, 42/174 for CLA$_{RSP}$ neurons). Similar to our single opsin observations, CLA$_{RSP}$ neurons were more likely to be responsive to inputs from frontal areas than they were from other areas, although at least some neurons were found to respond to both inputs in all examined pairs (*Figure 4D*). Integration was most common between ACAa and ORB (60.7%) and lowest between VISam and AUDd (16.7%). Less integration was observed when only one or neither of the input cortices was located in the frontal cortex. The measured probability of integration, however, was slightly higher than expected (ratio of measured:expected = 1.26 +- 0.12) based on the probability of receiving inputs from each cortical area individually, indicating that integration among single CLA neurons in these experiments occurred at a likelihood greater than response probabilities to individual cortical inputs would imply (*Figure 4E*).

Concerning the electrophysiological identities of CLA neurons themselves, only E2 and FS types received input from every cortical area (*Figure 4F*). Similarly, only ACA sent outputs to every CLA cell type, while AUDd sent outputs to just two (E2 and FS). IR cells were the only type found to receive input from only one area (ACA, n=6 cells; *Supplementary file 3*), although no IR neurons were

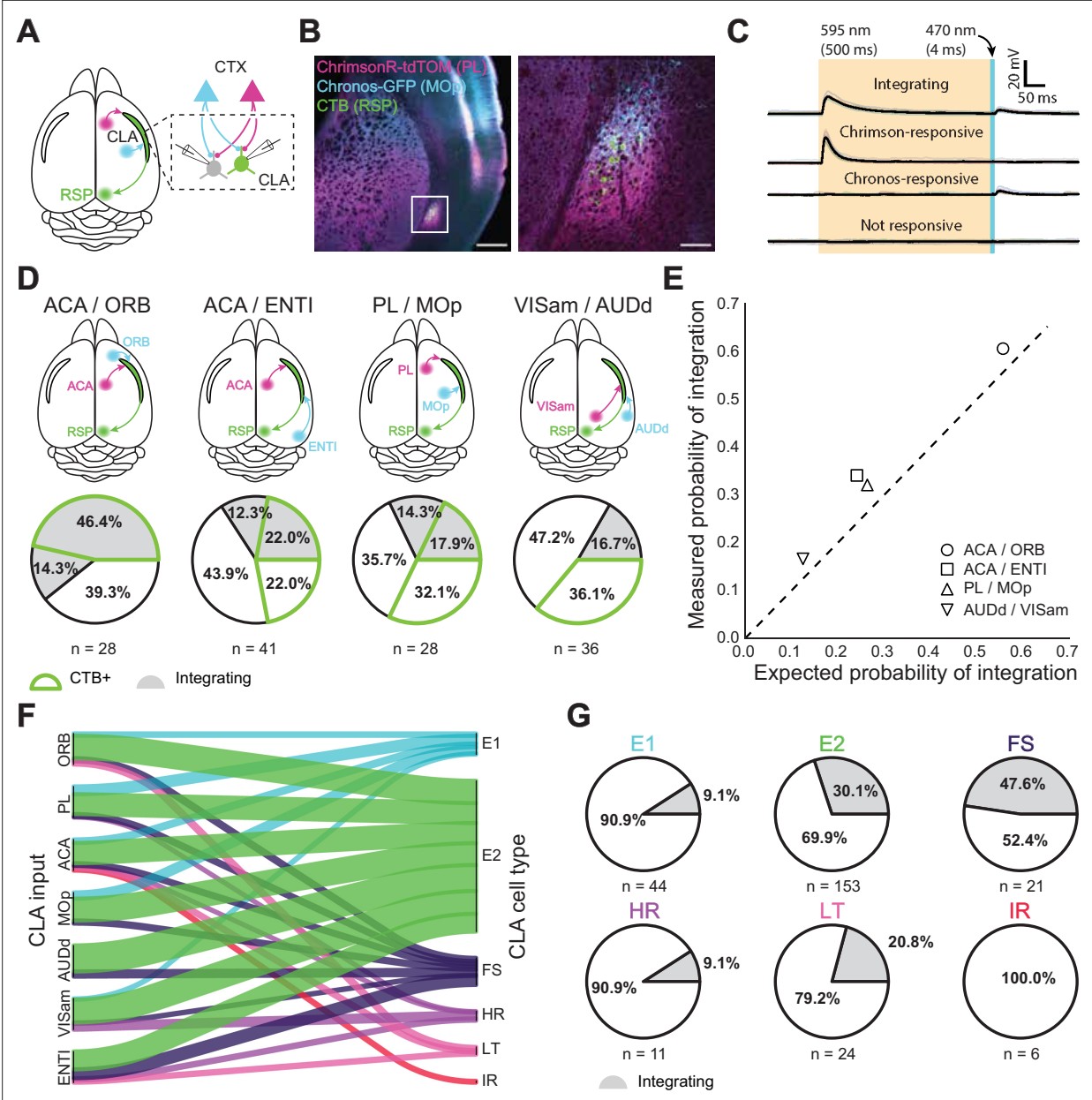

**Figure 4.** Dual-color optogenetics reveals integration among CLA neurons. (**A**) Schematic of injection and patching strategy in the CLA. (**B**) Confocal images of AAV-Chrimson-tdTomato and AAV-Chronos-GFP expression in cortical axons in the CLA. Scale bars = 500 µm, 100 µm. (**C**) Example traces of different response outcomes for CLA neurons after sequential cortical axon photostimulation. From top to bottom: integrating responses indicated that a neuron was responsive to both input cortices, Chrimson-responsive indicated a neuron responsive to only one cortex while Chronos-responsive indicated a neuron responsive to the other cortex. Finally, no response indicated no detectable synaptic connection. (**D**) Dual-color optogenetics response and CTB proportions for cortices examined in *Figure 3* using the strategy in (**C**). (**E**) Expected vs. measured response probabilities for neurons in each dual-color optogenetics combination. All cortical combinations displayed a slightly higher response probability than expected by chance. (**F**) River plot displaying the proportion of neurons projected to by each cortical area, categorized by cell type. (**G**) Proportion of integrating cells within each electrophysiological cell type.

The online version of this article includes the following figure supplement(s) for figure 4:

**Figure supplement 1.** Single-wavelength controls of dual-color optogenetics experiments.

recorded in experiments testing inputs from PL, MOp, AUDd, or VISam (sampling bias also affects the HR class, from which only 11 cells were recorded in optogenetic experiments). Overall, both excitatory classes were differentially innervated by the cortex. For example, 73% of all VISam inputs were to E2 neurons, while only 4% were allocated to E1 neurons. By contrast, 20% of MOp inputs and 25% of PL inputs were devoted to E1 neurons, while comparatively fewer CLA inhibitory types received inputs from these regions (10% and 11% in total to inhibitory neurons, respectively) than from ENTl (cumulatively 43% of inputs).

Integration of cortical inputs was more prevalent in certain cell types as well (*Figure 4G*). 30.1% of E2 neurons and 47.6% of all recorded FS interneurons were found to integrate cortical input, irrespective of the combination of cortical input regions, congruent with their response probabilities to individual cortices (*Figure 4F*). A large proportion (20.8%) of LT neurons were also found to integrate despite making up less than 10% of all neurons recorded. E1, HR, and IR types showed little to no propensity for more than one cortical input. We find that E2 neurons and FS interneurons are the most likely to integrate information from the cortex, while other excitatory and inhibitory cell types may participate in different circuits or have a dedicated and unitary region of input.

## Claustrocortical outputs differentially innervate cortical layers in downstream targets

To explore how CLA influences the neocortex, we returned to our retro-Cre conditional expression of opsin in $CLA_{RSP}$ neurons (*Figure 5A*), focusing on outputs to ACA and RSP as CLA connectivity with these areas is particularly strong (*Figure 5B*). Axonal fluorescence from CLA neurons varied by cortical layer in these regions (*Figure 5C*). Cells in ACA or RSP were filled with biocytin during recording for post hoc analysis of their location within the cortical laminae (*Figure 5D*). Optogenetic stimulation of CLA axons evoked both inhibitory postsynaptic currents (IPSCs) and excitatory postsynaptic currents (EPSCs) in cortical neurons during voltage-clamp at holding potentials of 0 mV and –70 mV, respectively (*Figure 5E*). Similarly to other experiments, we considered monosynaptic excitatory connections to be those with latencies of 3–12ms, here confirmed with pharmacological controls (*Figure 5F–J*, n=9 cells). The longer latency to onset of IPSCs (7–15ms) suggests recruitment of feed-forward inhibition by CLA neurons in both cortical areas, although some short-latency IPSCs could be due to direct long-range inhibitory projections (*Figure 1D*, *Figure 1—figure supplements 6 and 7*). However, no direct IPSCs were found after application of TTX and 4AP in control experiments (*Figure 5G and H*). PSCs could be evoked relatively evenly across most cortical layers of ACA (n=49 cells, *Figure 5K*, *top*). In RSP, we observed the highest response probability in L5, but responses in deep layers overall were reduced compared to ACA (n=43 cells, *Figure 5K*, *bottom*). Finally, we found that excitation and inhibition latencies were statistically different in ACA but not RSP (ACA p=0.0003, RSP p=0.057, Cochran–Mantel–Haenszel test). These experiments point to a complex interaction with target cortical areas that are both cortical area and layer-dependent.

## CLA axons respond to unimodal and multimodal stimuli during in vivo calcium imaging

The sum of our in vitro experiments points to CLA having a role in responding to and potentially combining inputs from higher order association areas rather than direct sensory binding. To explore the latter further, we next sought to understand if CLA signals sensory information to the cortex in vivo. We injected mice with a retro-Cre virus in ACA and RSP to maximize CLA labeling and a Cre-dependent calcium indicator (AAV-FLEX-GCaMP7b) in the CLA. Mice were subsequently implanted with cranial windows centered above bregma to capture midline-traveling CLA axons for observation during two-photon calcium imaging (78 recordings from 4 animals including 1364 axon segments; *Figure 6A*, *left*). Congruent with previous experiments, the expression of GCaMP7b was restricted to CLA neurons, and axons from these neurons were visible in the cortex (*Figure 6B*, *Figure 6—figure supplement 1*). GCaMP7b-labeled axons were recorded throughout the cranial window in the hemisphere ipsilateral to the injection.

To test our in vitro results demonstrating at least some sensory responsiveness in single CLA neurons, mice were exposed to stimuli intended to evoke responses in different sensory modalities: a flash of light, stimulation of the whisker pad via a piezo-controlled paddle, and/or a complex auditory tone (*Figure 6A*, *right*). We investigated sensory responses here to account for the discrepancy in our

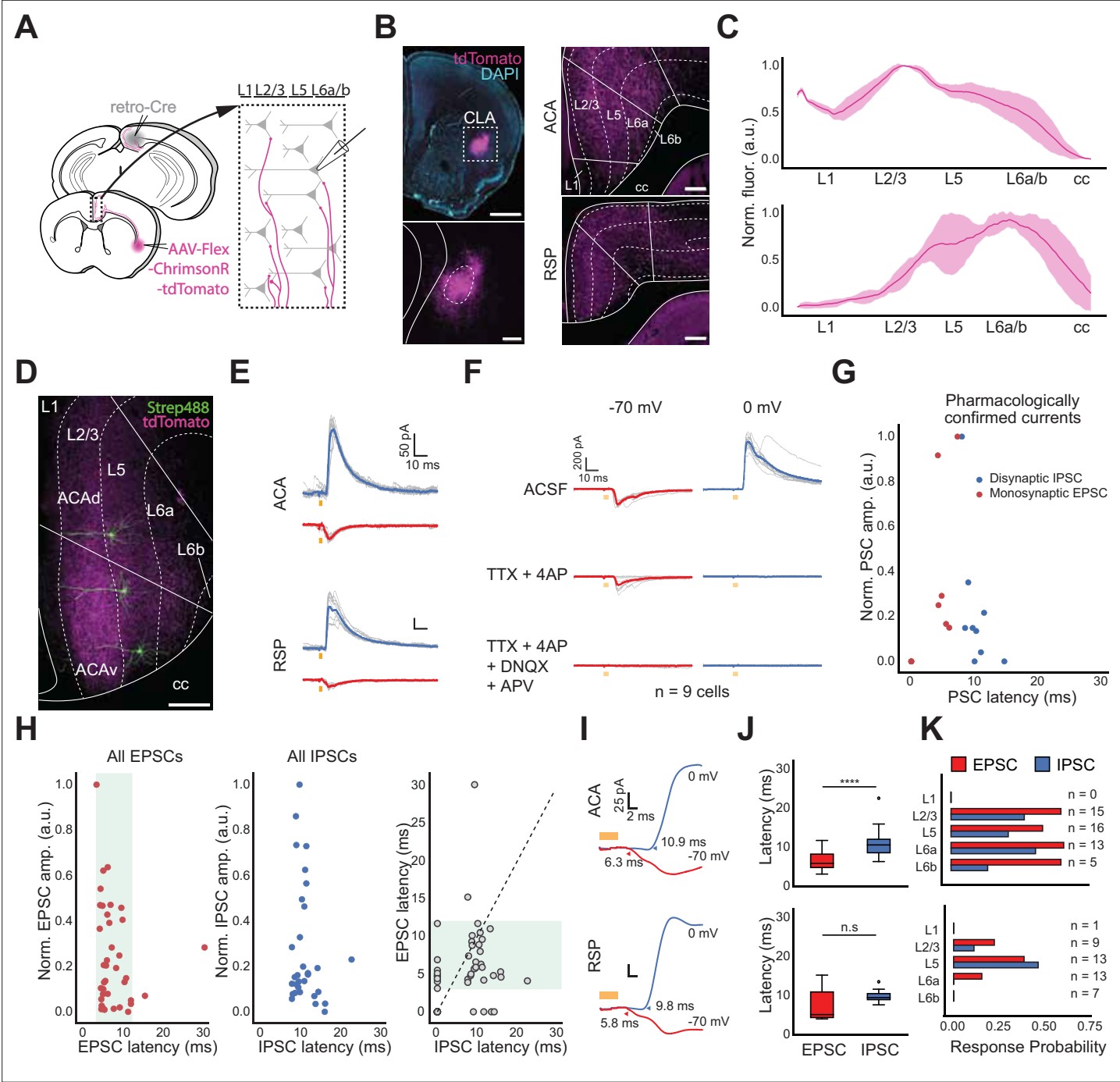

**Figure 5.** In vitro measurements of CLA afferents in cortex uncover layer specificity. (**A**, top) Schematic of injection and patching strategy for assessing cortex responses to photostimulation of CLA axons. (**B**, left) Representative images of opsin expression in CLA cell bodies. Scale bars = 1 mm (top), 200 µm (bottom). (**B**, right) Opsin expression in CLA axons innervating ACA (top) and RSP (bottom). Scale bars = 200 µm. (**C**) Normalized CLA axonal fluorescence in ACA and RSP. (**D**) Example image of biocytin-filled neurons in ACA. Scale bar = 200 µm. (**E**) Example average traces of IPSC (0 mV, blue trace) and EPSC (–70 mV, red trace) responses from a single neuron in both the ACA and RSP, aligned to light onset. (**F**) Pharmacological investigations of EPSC and IPSC responses to photo stimulation in normal ACSF (top) and during bath application of TTX and 4AP (middle) or TTX, 4AP, DNQX, and APV (bottom; n=9 cells, 3 mice). (**G**) Quantification of normalized PSC magnitude and latency in pharmacological experiments. (**H**) Same as in (**G**) with neurons recorded solely in ACSF (n=92 cells, 10 mice). (**I**) Expanded visualizations of currents in **E** demonstrating the differences in EPSC and IPSC latency in both ACA and RSP. (**J**) Quantification of EPSC and IPSC latency in ACA and RSP (ACA p=0.0003, RSP p=0.057, Cochran–Mantel–Haenszel test). (**K**) PSC probability in cortical neurons sorted by the layer in which neurons were patched.

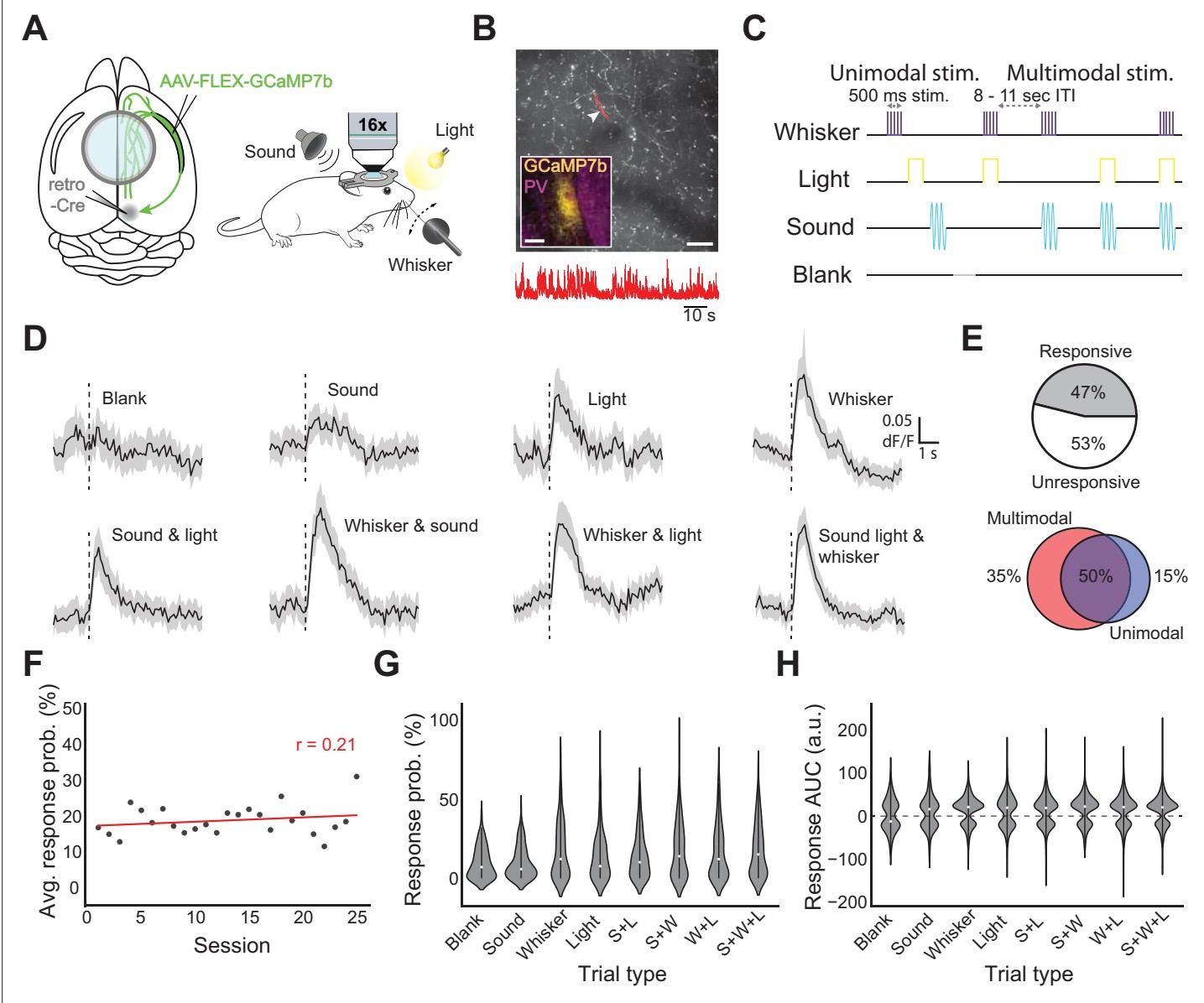

**Figure 6.** In vivo responses of CLA axons in cortex to sensory stimulation. (**A**, left) Schematic of injection strategy and window placement over bregma. (**A**, right) Schematic of in vivo recording strategy with symbols for stimuli (upper left: complex tone, upper right: white LED light, lower right: whisker stimulator). (**B**) Example FOV with CLA axons expressing GCaMP7b. Highlighted area and arrowhead indicate the axon from which the trace below was recorded. Scale bar = 50 µm, 10 seconds. Inset: GCaMP7b expression in CLA from approximately 0.0 mm bregma. Inset scale bar = 200 µm. (**C**) Passive stimulation protocol using three stimulus modalities. Stimuli and combinations thereof were presented 8–11 s apart (randomized) with a "blank" period where no stimulus was presented every ~8 trials. (**D**) Average dF/F traces for each stimulus and combination across all axons responsive to that modality. Black line indicates the population mean, grey shaded regions indicate the 95% CI. (**E**, top left) Proportion of all recorded axons displaying significant responses to one or more trial types (n=4 mice, 1364 axons, 78 recordings). (**E**, top right) Proportion of all responsive axons displaying a uni- or multisensory response pattern. (**E**, bottom) Some axons were modulated only by multimodal trial types (left), both unimodal and multimodal trial types (center), or only unimodal trial types (right). (**F**) Dots represent the probability of observing a sensory evoked response to any trial type averaged across all axons, FOVs, and mice in an imaging session. Pearson's r (0.21) indicates that response probability was not strongly correlated with experimental session. (**G**) Violin plots displaying the trial-by-trial response probability of axons to different trial types. (**H**) Violin plots displaying the trial-by-trial response magnitude of axons to different trial types.

The online version of this article includes the following figure supplement(s) for figure 6:

**Figure supplement 1.** Histological verification of indicator expression in the CLA of behavior mice.

**Figure supplement 2.** Population axonal responses to single and multimodal stimuli.

**Figure supplement 3.** In vivo axonal responses to single and multimodal stimuli.

results - which found strong frontal integration but relatively weak sensory integration despite strong visual input, with other reports of direct sensory responses in the CLA *Remedios et al., 2010*; *Olson and Graybiel, 1980*; *Sherk and LeVay, 1981*; *Alloway et al., 2009*. Therefore, we determined that it was necessary to investigate sensory-related activity in the CLA as a basis for modality-dependent integration.

We defined a stimulus-evoked response as any significantly large deflection during the one second post-stimulus presentation compared to one second before, corrected for multiple comparisons (see Materials and methods). Stimuli were randomized at 8–11 s intervals and interleaved with a 'blank' period in which no stimulus was delivered. Trials were either unimodal or multimodal: either one stimulus was presented alone, or more than one was presented simultaneously (*Figure 6C*). 47% of tested axons displayed significant calcium transients to at least one stimulus modality during passive presentation, and all modalities could evoke responses in at least some CLA axons (*Figure 6D and E* top left; *Figure 6—figure supplement 2D*, right). For unimodal stimulus presentations, somatosensory stimuli (whisker) were the most likely to elicit changes in fluorescence, followed by light and then sound (18.8% of recorded axons responded to whisker, 12.1% to light, 10.9% to sound). For pairs of stimuli, the combination of somatosensory and visual stimuli drove the largest changes in fluorescence (19.1%), followed by somatosensory and auditory (15.8%), then auditory and visual (14.1%). The combination of all three stimuli resulted in the largest proportion of activated axons (20.7%), while the blank period was associated with the fewest responsive axons (9.8%, *Figure 6—figure supplement 3*). 85% of responsive axons were responsive to at least one or more multi-modal trial types. Each axon was then classified as either uni- or multisensory based on the modalities present in the trial types to which they responded (*Figure 6E top right*). Of sensory-responsive axons, only 4% of axons were found to display unisensory response patterns, while 96% displayed multisensory response patterns. We then examined the response to unimodal and multimodal stimuli irrespective of the modalities presented. Interestingly, 35% of stimulus-responsive CLA axons were exclusively responsive to multimodal trial types, while 15% were exclusively responsive to unimodal trial types (*Figure 6E, bottom*). Experiments were repeated using only unimodal stimuli (i.e. sound, light, and whisker only), and similar results were obtained (117 recordings from 4 animals including 1342 axon segments; *Figure 6—figure supplement 2*).

To understand the trial-to-trial diversity of axonal responses, we examined the post-stimulus AUC for the dF/F of each axon. With respect to the reliability of axonal responses to sensory stimulation, we found that the probability of observing a sensory-evoked response in a given field of view regardless of stimulus type did not change, on average, over the course of experimentation (*Figure 6F*, $r$=0.21, $p$=0.32). However, axonal response probability was significantly modulated between stimulus types (*Figure 6G*, $p$=3.7e-9 Kruskal Wallis test). Similarly, stimulus type was found to be a significant source of variation in the magnitude of axonal responses (*Figure 6H*, $p$=6.4e-10 Kruskal Wallis test).

These data are consistent with our in vitro recordings that suggested frontal cortical input integration among CLA neurons was a common occurrence and/or that CLA neurons receive input from a cortical region that contains neurons of mixed selectivity (*Figure 4*). We also find that CLA axonal responses to passive sensory stimulation are durable across recording sessions and between stimulus modalities. The results above collectively indicate that CLA outputs to the cortex convey higher order information that likely arises from integration of either weak and direct sensory input from primary cortices or elicited indirectly via integration of input from frontal association cortices.

## CLA silencing reduces sensitivity to multimodal stimuli

After observing the multimodal properties of the $CLA_{RSP}$ both in vitro and in vivo, we next sought to examine the functional relevance of the $CLA_{RSP}$. To understand the contribution of $CLA_{RSP}$ to behavior, we chronically silenced $CLA_{RSP}$ output to the cortex using virally expressed Tetanus Toxin Light Chain (TetTox). We first validated the effectiveness of TetTox-based $CLA_{RSP}$ silencing using an optogenetic approach in acute in vitro slices of cortex (*Figure 7A*). Patch-clamp recordings of RSP neurons revealed that TetTox reduced the frequency and magnitude of light-evoked currents in downstream cells, effectively silencing $CLA_{RSP}$ output ($p$=0.0003, *Figure 7B and C*).

We assessed the effect of $CLA_{RSP}$ silencing across an array of behavioral assays (*Figure 7D*, *Figure 7—figure supplements 1–3*, see Methods). First, we compared animals injected bilaterally with AAV-retro-iCre-mCherry in RSP and AAV-FLEX-TetTox in CLA (TetTox group) with animals injected

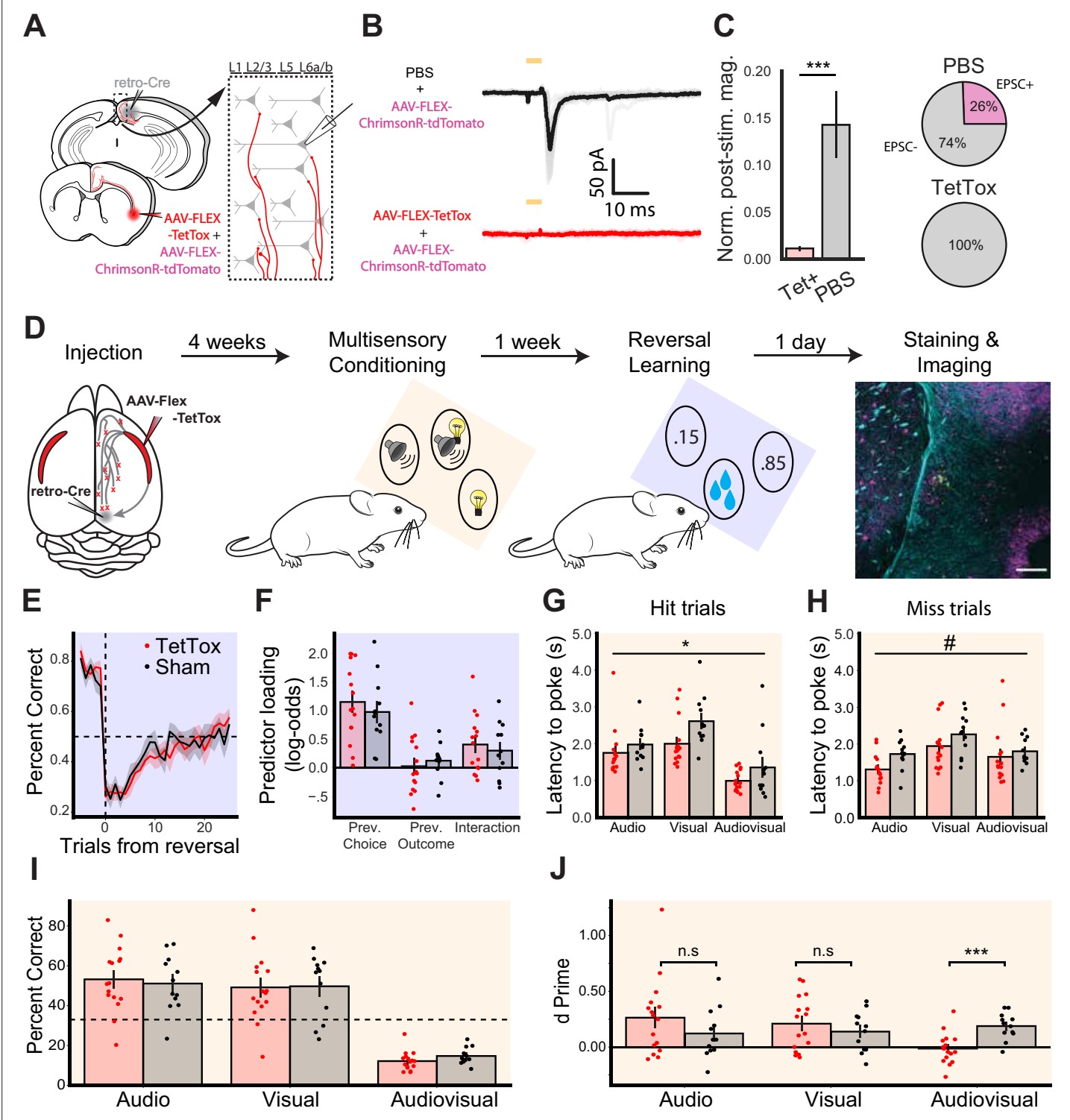

**Figure 7.** CLA silencing reduces sensitivity to multimodal stimuli. (**A**) Schematic of retrograde injection and patching strategy for assessing silencing of CLA$_{RSP}$ terminals in cortex. (**B**) Example average voltage clamp electrophysiology from RSP neurons in mice injected with PBS and AAV-ChrimsonR-tdTomato (top, n=33 cells) or AAV-FLEX-TetTox and ChrimsonR (bottom, n=16 cells, 2 mice), aligned to light onset. (**C**, left) Quantification of normalized PSC magnitude in RSP neurons in Tet+ (red) and PBS (gray) conditions. (Right) Percent of neurons in each injection condition showing EPSCs in response to CLA$_{RSP}$ axon photostimulation. (**D**) Experimental pipeline for behavioral experiments, beginning with injection and finishing with histological verification. Plots shaded blue correspond to the reversal learning task, while plots shaded orange correspond to the multimodal conditioning task. (**E**) Average probability of poking the high reward probability port in the trials before and after the transition to a new block (trial 0). Dashed lines

*Figure 7 continued on next page*

*Figure 7 continued*

indicate chance performance and block transition (two-way repeated measures ANOVA, effect of group $F_{(1,25)}$ = 0.413, p=0.526). (**F**) Loading of logistic regression predictors based on data from the final five sessions (Welch's t-test for previous choice p=0.482, previous outcome p=0.539, and interaction p=0.581). In the multimodal conditioning task, two-way repeated measures ANOVA was used to compare the latency from stimulus onset until the first poke, regardless of which port was poked, on hit (**G**; $F_{(1,26)}$ = 6.252, p=0.019) and miss (**H**; $F_{(1,26)}$ = 3.527, p=0.072) trials. (**I**) Percentage of trials classified as hits for each trial type (two-way repeated measures ANOVA, effect of group $F_{(1,26)}$ = 0.056, p=0.814). Dashed line indicates chance performance of 33%. (**J**) d' values calculated separately for each stimulus type. Multimodal conditioning plots include data from experienced mice only (training sessions after day 3). Data from sham mice (n=12) are plotted in black, and TetTox mice (n=16) are plotted in red. Error bars show the standard error of the mean. Symbols indicate an effect of treatment where p≤0.1 (#), p≤0.05 (*), or p≤0.001 (***).

The online version of this article includes the following figure supplement(s) for figure 7:

**Figure supplement 1.** Immunofluorescent staining of TetTox injected (CLA) brain tissue showing parvalbumin (PV, magenta, top), myelin basic protein (MBP, cyan, top center), retro-iCre-mCherry (CRE, red, center), FLAG (yellow, bottom center), and overlay (bottom).

**Figure supplement 2.** Immunofluorescent staining of TetTox injected brain tissue (**A**) and PBS injected brain tissue (**B**) showing glial fibrillary acidic protein (GFAP; top), myelin basic protein (MBP; middle), and overlay (bottom).

**Figure supplement 3.** No change due to claustrum silencing in 24/7 activity levels nor ethologically motivated anxiety assays.

**Figure supplement 4.** Reversal learning task.

**Figure supplement 5.** Multimodal conditioning task.

with equal volumes of PBS (sham group) during 24/7 home cage recordings using measures of activity, circadian behavior, and tests for anxiety. We found that silencing $CLA_{RSP}$ output did not significantly alter the behavior of TetTox animals compared to the sham group. Post hoc histology from these groups revealed increased expression of glial fibrillary acidic protein (GFAP) in the CLA and reduced mCherry expression, strongly suggesting full ablation of $CLA_{RSP}$ neurons in these experiments.

We next tested whether the effects of $CLA_{RSP}$ axon silencing would be more apparent in a complex sensory or cognitive behavioral paradigm. We trained a cohort of mice with chronic and sham $CLA_{RSP}$ silencing on complex behavioral tasks: a multimodal conditioning task and a reversal learning task (*Figure 7D–J*, see Materials and methods). In the reversal learning task, mice learned to choose between two nose-poke ports associated with different probabilities (85% and 15% chance, switched after learning the task to threshold). We identified no differences between CLA-silenced and sham groups across a number of metrics including response probability and latency (*Figure 7—figure supplement 4*). The results from the reversal learning task provide no evidence that CLA silencing affects learning or cognitive flexibility.

Drawing from our calcium imaging results described above (*Figure 6*), we further assessed how mice learned to link their actions to patterns of uni- and multimodal stimuli in their environment using a multimodal conditioning task. Water-restricted mice were presented with auditory, visual, and combined audiovisual stimuli. Each of the three stimuli was associated with reward delivery at a different nose-poke port, with the reward delivered in response to the first poke to the correct port following stimulus onset, irrespective of whether the animal had initially poked an incorrect port. While mice from both groups learned the structure of the task (*Figure 7G*; *Figure 7—figure supplement 5*), two differences were detected between sham and $CLA_{RSP}$ silenced mice. First, we found that $CLA_{RSP}$-silenced mice had significantly lower poke latencies across trial types and outcomes (*Figure 7H,I*). Second, a two-way ANOVA identified a significant interaction when comparing animals' sensitivity (the difference in the distributions of the hit rate and false alarm rate) to different trial types ($F_{(2, 52)}$=5.53, p=0.006; *Figure 7J*). After correcting for multiple comparisons (Sidak), we found that $CLA_{RSP}$ silencing specifically impaired animals' sensitivity to multimodal stimuli (p=0.001). These results suggest that $CLA_{RSP}$ silencing specifically reduced animals' sensitivity to multimodal stimuli without impacting their responses to unimodal stimuli.

Collectively, these results suggest a nuanced and specific function for the CLA. We found no effect of $CLA_{RSP}$ silencing on learning or cognitive flexibility in the reversal learning task. However, we observed that $CLA_{RSP}$ silencing decreased animals' poke latency and sensitivity to multimodal stimuli. This suggests that the integrative properties of the CLA described above may have functional relevance for detecting the coincidence or conjunction of two stimuli in the context of sensory-guided behavior.

# Discussion

The extensive connectivity of the CLA suggests a multifaceted role within the brain. In this study, we exploited the specificity of CLA projections to RSP to explore the electrophysiological diversity of CLA neurons both within the CLA$_{RSP}$ module and broader CLA. Further, we used a range of optogenetic and imaging approaches to assess their role in corticoclaustral, intraclaustral, and claustrocortical circuits. Our results show that individual CLA neurons integrate diverse information from across the cortex, that is are responsive in vitro and in vivo to a broad spectrum of cortical input modalities, in a cell-type-specific manner. We find a robust intraclaustral network of excitatory neurons that are differentially responsive to combinations of cortical input based on cell type. In addition, we show that CLA$_{RSP}$ neurons innervate the cortex in a region- and layer-specific manner and, when silenced, specifically impair behavioral performance during a multimodal conditioning task.

Our study builds on previous work to label a specific subset of CLA neurons by using a retrograde tracing strategy that co-opted the highly specific connectivity of the CLA with RSP (*Zingg et al., 2018*; *Marriott et al., 2021*; *Erwin et al., 2021*). RSP is uniquely positioned for use in this technique as it does not receive inputs from structures around the CLA but receives specifically dense innervation from the CLA itself. We found that CLA$_{RSP}$ neurons span the rostrocaudal axis of the CLA and align with previously identified markers of the CLA 'core' (*Druga et al., 1993*; *Real et al., 2003*; *Real et al., 2006*; *Grimstvedt et al., 2022*). For these reasons, we found this method to be favorable over transgenic or viral labeling used in other studies (reviewed in *Jackson et al., 2020*; see also *Wang et al., 2023*; *Chia et al., 2020*; *Chevée et al., 2022*; *McBride et al., 2023*; *Ollerenshaw et al., 2021*). Retrograde tracing from RSP offered a simple method for accurately assessing the anatomical and physiological features of CLA neurons in later experiments.

Retrograde labeling of CLA$_{RSP}$ proved useful for targeted investigations of CLA electrophysiology in vitro (*McGarry et al., 2010*; *Toledo-Rodriguez et al., 2004*). From our recordings across a large population of both CLA$_{RSP}$ and non-CLA$_{RSP}$ neurons, it was evident that a heterogeneous mix of spiny, excitatory neurons and aspiny, inhibitory neurons exists in the CLA, consistent with other studies (*Graf et al., 2020*; *Druga, 2014*; *Spahn and Braak, 1985*). These broad categories could be divided into subgroups consisting of six total electrophysiological types, two excitatory and four inhibitory. Similar to adjacent neocortex (*Gouwens et al., 2019*), excitatory neurons were homogenous from an unsupervised clustering perspective but could nevertheless be differentiated by their AP waveforms and variability in their tendency to project to RSP. Although direct comparisons are difficult, the E1, E2, FS, and LT subtypes closely matched excitatory and inhibitory cell types found in recent investigations of CLA intrinsic electrophysiology (*Graf et al., 2020*; *Kim et al., 2016*; *Qadir et al., 2022*). For example, E1 neurons share several characteristics with Type I neurons described by *Qadir et al., 2022*, including monophasic AP amplitude adaptation, while E2 neurons strongly resemble Type II neurons, specifically in their tendency to fire a burst of strongly adapting spikes.

Inhibitory neurons, by contrast, were accurately distinguished using unsupervised methods and are similar to those observed in the neocortex (*Rudy et al., 2011*). In addition, we found multiple lines of evidence indicating the existence of a substantial subpopulation of inhibitory projection neurons, which have also been observed in prefrontal cortex, amygdala, hippocampal areas, entorhinal cortex, and the subplate (*Melzer et al., 2012*; *Basu et al., 2016*; *Boon et al., 2019*; *Jinno et al., 2007*; *Lee et al., 2014*; *Melzer and Monyer, 2020*; *Molnár and Butler, 2002*), with additional evidence of such cells indicated in previous studies of the CLA (*Kitanishi and Matsuo, 2017*; *Atlan et al., 2018*). These findings not only highlight the similarity of CLA to other forebrain structures (*Bruguier et al., 2020*) but also suggest previously unconsidered functional possibilities. The putative monosynaptic inhibitory inputs may provide another route by which CLA exerts a direct suppressive influence on the cortex.

Hypotheses that position the CLA as affecting cross-modal processing (*Calvert, 2001*; *Ettlinger and Wilson, 1990*), synchronization *Smythies et al., 2012*; *Smythies et al., 2014*, or multimodal integration (*Crick and Koch, 2005*; *Vidyasagar and Levichkina, 2019*) implicitly rely on a substantive intraclaustral excitatory network to link projection neurons across its considerable length. Here, we used a dual-retrograde and conditional opsin expression strategy to understand whether such connections are present in the CLA, as has been debated elsewhere (*Zingg et al., 2018*; *Kim et al., 2016*; *Orman, 2015*; *Smith and Alloway, 2014*; *LeVay, 1986*). Importantly, our optogenetic stimulation approach (*Petreanu et al., 2007*) allowed us to remain agnostic to the origin of presynaptic signals in

the CLA – a key improvement over paired-patching, which is susceptible to slice-orientation artifacts (e.g. coronal vs. horizontal sectioning). We found that excitatory connections are quite common in the CLA and broadly target most CLA excitatory and inhibitory types. Additionally, we observed that this connectivity was less biased toward inhibitory types than previously thought (*Kim et al., 2016*), but was influenced more by the output target of postsynaptic CLA neurons and the axis along which excitatory connectivity predominantly acts, that is rostrocaudally. Specifically, we observed a difference in the likelihood of excitatory signaling between CLA neurons that was dependent on whether neurons were retrogradely labeled by their projections to RSP or PL. CLA$_{PL}$ and non-CLA$_{RSP}$ neurons were both more likely to receive input from CLA$_{RSP}$ than CLA$_{RSP}$ neurons themselves. Combined with our later finding that PL preferentially targets CLA$_{RSP}$, these results provide substantial evidence for cross-modular communication between corticoclaustral input streams.

Much like CLA efferents (*Marriott et al., 2021*), we found that cortical projections to CLA arrange into modules along the dorsoventral axis, a similar finding to other studies (*Atlan et al., 2017*; *Wang et al., 2023*; *Wang et al., 2017*). Interestingly, certain cortices such as ORB and ACAa projected to CLA$_{RSP}$ both medially and laterally in addition to dorsally or ventrally. Physiological investigations of cortical input revealed that CLA$_{RSP}$ neurons are more likely to respond to frontal cortical regions than non-CLA$_{RSP}$ neurons. CLA$_{RSP}$ neurons were also more likely than non-CLA$_{RSP}$ neurons to respond to motor and association cortices, despite the stark regionalization of both cortical axons and postsynaptic responses in the CLA. Surprisingly, however, we found that CLA neurons, especially non-CLA$_{RSP}$ neurons, were far more likely to respond to secondary visual cortex input in vitro, more so than has been reported in primary sensory cortices (*Chia et al., 2020*; *White et al., 2018*). However, we could not assess and compare the absolute response magnitudes to these inputs due to confounds presented by opsin-mediated presynaptic release – differential expression of opsin in presynaptic neurons or between animals could result in erroneous estimates of naturally evoked EPSP amplitudes. Despite this, these findings both confirm the deep ties between CLA and frontal areas associated with top-down cognitive functions and suggest higher responsiveness to more highly processed sensory information.

The patterning of cortical axons in the CLA of mice is simultaneously segmented, with identifiable dorsal, central, and ventral modules, while also forming an overlapping gradient (*Atlan et al., 2017*; *Qadir et al., 2022*; *Wang et al., 2017*) that blends input streams to CLA neurons. From an anatomical perspective, we thought it very likely that CLA neurons in mice instantiate multimodal integration at the level of single cells given the overlap of cortical afferents within it, despite previous reports of unisensory modules in the CLA of cats and monkeys (*Remedios et al., 2010*; *LeVay and Sherk, 1981*; *Olson and Graybiel, 1980*). To test whether this was the case, we used a dual-color optogenetic input mapping strategy to assess the responsiveness of CLA$_{RSP}$ neurons and non-CLA$_{RSP}$ neurons to more than one cortical area in vitro (*Yuan et al., 2015*). As a necessary constraint of our photostimulation paradigm, we did not assess the EPSP response magnitude to the conjunction of stimuli due to photosensitivity of ChrimsonR opsins to blue light (*Klapoetke et al., 2014*). Our findings demonstrate that individual CLA neurons are frequently responsive to multiple different inputs. This was especially true for CLA$_{RSP}$ when the cortices in question were both frontal, while the balance of responsiveness shifted to non-CLA$_{RSP}$ when other cortical areas were involved. Specifically, we found the E2 and FS CLA cell types to be highly integrative, while other types were much less so. In addition, E2 excitatory neurons were the most likely to project to RSP. Given that RSP is strongly associated with contextual and spatial awareness (*Trask et al., 2021*), it is possible that integration of inputs to the CLA by E2 neurons contributes to contextual processing in the RSP and may explain our multimodal conditioning results when these neurons were silenced. Overall, cell type identity, as defined by intrinsic electrophysiology and efferent projection target, influenced the likelihood that a neuron was dual-responsive to afferent input in the CLA in these experiments.

In vitro cortical optogenetic experiments sought to investigate regional and laminar differences in CLA innervation of the cortex. Recent in vivo electrophysiological evidence points toward differences in excitatory and inhibitory tone elicited by excitation of CLA cell bodies that varies by cortical area and layer (*Atlan et al., 2021*; *McBride et al., 2023*). Other studies in single cortical regions find more uniform responses to CLA inputs, generally inhibitory (*Jackson et al., 2018*; *Atlan et al., 2018*), although electrophysiological studies in cats have found more variable or bidirectional responses in visual cortices (*Cortimiglia et al., 1991*; *Tsumoto and Suda, 1982*). We chose to investigate CLA$_{RSP}$

connections to ACA and RSP in vitro and in vivo – ACA for the dense connectivity it shares with CLA (and CLA$_{RSP}$) and RSP for the known properties of neurons that project there. We found that CLA axons innervate the cortical layers of ACA and RSP differently, confirming and expanding on results from recent works (*Jackson et al., 2018*; *McBride et al., 2023*). Overall, ACA was innervated relatively evenly across layers and more so in deep layers than RSP for excitation and inhibition. It is possible, given this, that the cortex-dependent laminar innervation by CLA axons reflects differences in the types of information conveyed to the cortex by CLA neurons or, perhaps, the types of cortical neurons they form synapses with and the dendritic locations on which those synapses occur (*McBride et al., 2023*; *Larkum et al., 1999*).

Our in vivo calcium imaging experiments identified a population of neurons within the CLA that showed stimulus-locked responses to presentations of visual, auditory, and tactile stimuli. A large proportion of responsive CLA axons were exclusively activated in trials where two or more stimuli were presented simultaneously. While axons responded inconsistently to the presentation of individual stimuli on a trial-to-trial basis, their responsivity remained consistent between stimuli and through time on average, implying that CLA may not be involved in habituation or adaptation of responses. While the observed responses coincide with sensory stimulation, they may not be sensory per se. We thought that these responses may instead be related to attention, salience, or motor processes occurring as a result of the sensory stimulation, motivating our use of somatosensory and auditory stimuli despite weak direct inputs from these cortices, documented elsewhere (*Remedios et al., 2010*; *Chevée et al., 2022*). The large proportion of activated axons contrasts with recent studies in mice *Chevée et al., 2022*; *Ollerenshaw et al., 2021* in which CLA neurons were infrequently responsive to sensory stimulation in vivo. The disparity between these results may be explained by methodological differences. Both studies used different methods of labeling and recording from CLA neurons: Ollerenshaw et al. used a transgenic line (Gnb4) and calcium imaging via a GRIN lens implant, while Chevée et al. optotagged neurons based on their projections to somatosensory cortex and recorded extracellular electrical activity. Both these approaches did not explicitly target the same neurons reported here (CLA$_{RSP}$), making comparison difficult. Moreover, the implanted devices used for recording likely resulted in some damage to local circuitry, affecting CLA activity, thus introducing a confound that would further impede one-to-one statistical comparisons. These techniques are also inherently limited in the region of CLA that is recordable at any given time. While axon imaging theoretically permits recording of axons originating throughout CLA, extracellular electrophysiology and implanted optical devices are limited to neurons in the vicinity of the recording device. As a result, each of these studies likely offers a different and possibly non-overlapping account of CLA activity.

Chronic claustrum silencing specifically reduced animals' sensitivity to multimodal, but not unimodal, stimuli. Previous studies have assessed the effects of both chronic and acute CLA silencing on reversal learning with mixed results (*Fodoulian et al., 2020*; *Grasby and Talk, 2013*; *Reus-García et al., 2021*), suggesting a nuanced role for the CLA in cognitive flexibility that can only be identified under specific behavioral constraints. Here, the mice tested on the reversal learning task were trained for 10 days, and as such, it is possible that we did not identify differences that might have occurred at later stages of learning. Importantly, the multimodal conditioning task did not force mice to learn an association between stimulus and port as the rewards were not contingent on choosing the correct port. The mice could have earned as many rewards by poking each port once per minute as they could through perfect task performance. Instead, this paradigm asked what information from the environment mice used to guide their behavior. The finding that sham mice could discriminate between unimodal and multimodal stimuli while CLA-silenced animals could not implies that CLA activity may be specifically related to responding to the conjunction of sensory stimuli. While this may provide evidence that the CLA is involved in detecting the coincidence of audiovisual stimuli, there is still debate about whether this constitutes multisensory processing (*Crick and Koch, 2005*; *Remedios et al., 2010*; *Chevée et al., 2022*; *Kim et al., 2016*; *Ollerenshaw et al., 2021*). Therefore, careful examination of CLA activity during multisensory-guided behavior will be necessary in future experiments.

It is possible that the integration of sensory inputs may take place upstream of the CLA rather than in the CLA itself. For example, the prefrontal cortex projects strongly onto CLA neurons and itself contains neurons responsive to sensory stimulation. Our in vitro results, which found few neurons responsive to the conjunction of sensory cortical input, may indicate that the appearance of sensory

integration in vivo could occur elsewhere. In vitro experimentation also does not measure the sensory responsiveness of CLA neurons but rather reveals solely direct, monosynaptic connectivity. Additionally, sensory responsiveness may come from other characteristics of the sensory stimulus such as their salience. It is also important to consider that CLA neurons might be responsive to combinations of stimuli, including unrecorded variables such as the animal's attentional or motivational states, that might affect their activity in vivo. Finally, different populations of CLA neurons not assessed here may be more responsive to and, therefore, responsible for the processing of direct sensory information in the CLA. Further work here is crucial to determine the nature of the sensory-related signals that the CLA routes to its cortical targets.

In summary, we find that CLA neurons are broadly capable of synthesizing a wide range of cortical inputs at the level of single neurons, a finding that supports the idea of CLA acting as a cortical network hub. The presence of a robust internal network of excitatory CLA neurons additionally supports the view that the CLA performs local computations. These computations then differentially influence downstream cortical processing in a regional- and layer-specific manner that may depend on the specific CLA output modules that are active at the time. Functionally, the fast monosynaptic connections investigated in this study could give rise to coincidence detection through these integrative and cross-modular CLA networks, providing all-or-none signals to the conjunction of stimuli as they arise in different sensory streams, as suggested elsewhere (*Chia et al., 2020*; *Kim et al., 2016*). CLA neurons may alternatively be involved in yoking together different streams of input based on their temporal synchrony, increasing their activity in a graded manner in response to more synchronous inputs (*Bruno, 2011*). For example, the internal, cross-modular, and recurrent excitation within the CLA could also allow for a degree of temporal synchronization between its input cortices through reverberant cortico-claustral loops (*Smythies et al., 2012*; *Vidyasagar and Levichkina, 2019*). Furthermore, the integrative properties of the CLA could act as a substrate for transforming the information content of its inputs (e.g. reducing trial-to-trial variability of responses to conjunctive stimuli and/or increasing conjunctive stimuli signal-to-noise). This would allow the CLA to flexibly modulate cortical activity, either through amplifying behaviorally relevant processes, diminishing irrelevant ones, or both (*Atlan et al., 2018*; *Murray and Wallace, 2011*).

The possible functions of CLA activity presented above point directly to and draw from its fundamentally integrative nature at the anatomical and functional levels. The findings shown here suggest that the CLA is involved in the highest levels of behavior, possessing the crucial neural substrates for a diverse and powerful effect on higher order brain function.

## Materials and methods

### Animals
Animal procedures were subject to local ethical approval under PPL #PE5B24716 and adhered to the United Kingdom Home Office (Scientific Procedures) Act of 1986. Male and female C57BL/6 J or Nkx2.1Cre;Ai9 mice were used in these experiments. Mice were between 3 and 11 weeks of age when surgery was performed. Long-Evans rats were used in experiments conducted at the Kavli Institute for Systems Neuroscience at the Norwegian University of Science and Technology (NTNU), Trondheim. These experiments were approved by the Federation of European Laboratory Animal Science Association (FELASA) and local authorities at NTNU.

### Stereotaxic surgery
Cortical and claustral injections of viruses and/or retrograde tracers were performed in mice aged p22–40. Briefly, mice were anesthetized under 5% isoflurane and placed in a stereotaxic frame before intraperitoneal injection of 5 mg/kg meloxicam and 0.1 mg/kg buprenorphine. Animals were then maintained on 1.5% isoflurane and warmed on a heating pad at 37 °C for the duration of the procedure. The scalp was sterilized with chlorhexidine gluconate and isopropyl alcohol (ChloraPrep). Local anesthetic (bupivacaine) was applied under the scalp two minutes before making the initial incision. The scalp was then incised along the midline and retracted to expose the skull, which was then manually leveled between bregma and lambda. Target regions were found using coordinates derived from the Paxinos & Franklin Mouse Brain Atlas (3rd ed.) and marked onto the skull manually (see *Table 1* for coordinates). Craniotomies were performed using a dental drill (500 µm tip) at 1–3 sites above the cortex.

Craniotomies were made exclusively in the right hemisphere unless otherwise noted. Pulled injection pipettes were beveled and back-filled with mineral oil before being loaded with one or more of the following: AAV1-Syn-ChrimsonR-tdTomato (Chrimson, 2.10e+13 gc/mL, 250 nL, Addgene #59171-AAV1), AAV5-Syn-FLEX-rc [ChrimsonR-tdTomato] (FLEX-Chrimson, 1.20e+13 gc/mL, 250 nL, Addgene #62723-AAV5), AAVrg-hSyn-Cre-WPRE-hGH (retro-Cre, 2.10e+13 gc/mL, 80 nL, Addgene #105553-AAVrg), AAV1-Syn-Chronos-GFP (Chronos, 2.90e+13 gc/mL, 250 nL, Addgene #59170-AAV1), AAV-syn-FLEX-jGCaMP7b-WPRE (FLEX-GCaMP7b, 1.90e+13 gc/mL, 250 nl, Addgene #104493-AAV1), ssAAV-retro/2-hSyn1-mCherry_iCre-WPRE-hGHp(A) (retro-iCre-mCherry, 5.00e+12 gc/mL, 80 nL, ETH Zurich VVF v230-retro 20740), ssAAV-DJ/2-hEF1a-dlox-FLAG_TeTxLC(rev)-dlox-WPRE-hGHp(A)(FLEX-TetTox, 6.810e+12 gc/mL, 500 nl, ETH Zurich VVF v63-DJ 20570), Cholera Toxin Subunit B (Recombinant) Alexa Fluor 488/555/647 Conjugate (CTB-488/555/647, 0.1% wt/vol, 80 nL, Thermo Fisher C34775/C34776/C34778, injected specifically into rostral, middle, and caudal RSP). Pipettes were lowered to the surface of the pia at the center of the craniotomy and zeroed before being lowered into the brain. In the case of injections into the CLA, specifically, the coordinates (*Table 1*) were intentionally offset in order to avoid the risk of damaging cells in that region with the pipette or by the injection of substances. The pipette was allowed to rest for 2 min before injection of substances, at which point injection took place at 5–10 nl/s. Pipettes were allowed to rest for ten minutes after injection. The incision was sutured with Vicryl sutures and sealed with Vetbond (3 M) after all craniotomies and injections had been made. Mice were then transferred to a fresh cage and allowed to recover. Mice were supplied with edible meloxicam jelly during post-op recovery for additional analgesia.

Mice to be implanted with cranial windows first received intracranial injections as described above. Once fully recovered from the injection surgery, mice were re-anesthetized for window implantation. Surgical preparation, anesthesia, analgesia, and recovery procedures were the same as for intracranial injection surgeries. Following sterilization of the scalp, a section was removed. The skull was then cleaned to remove the periosteum. An aluminum headplate with an imaging well centered on bregma was then secured in place with dental cement (Super-Bond C&B, Sun-Medical). A 4 mm circular craniotomy centered on bregma was then drilled. After soaking in saline, the skull within the craniotomy was removed. The craniectomy was then flushed with sterile saline to clean any bleeding. A durotomy was then performed over the right hemisphere. A cranial window composed of a 4 mm circular coverslip glued to a 5 mm circular coverslip was pressed into the craniotomy and sealed with cyanoacrylate (VetBond) and dental cement. Mice were then allowed to recover fully before any further experimental procedures.

## In vitro slice preparation

Acute coronal brain slices (300 μm thick) were prepared from tracer- and/or virus-injected mice (average age at time of experimentation = p52). Slices from virus-injected mice were prepared exclusively 3–5 weeks post-injection. Mice were deeply anesthetized with 5% isoflurane and transcardially perfused with ice-cold NMDG ACSF of the following composition: 92 mM N-Methyl-D-Glucamine (NMDG), 2.5 mM KCl, 1.25 mM NaH$_2$PO$_4$, 30 mM NaHCO$_3$, 20 mM HEPES, 25 mM glucose, 2 mM thiourea, 5 mM Na-ascorbate, 3 mM Na-pyruvate, 0.5 mM CaCl$_2$·4H$_2$O and 10 mM MgSO$_4$·7H$_2$O, 12 mM N-acetyl-cysteine (NAC), titrated pH to 7.3–7.4 with concentrated hydrochloric acid, 300–310 mOsm. The brain was then extracted, mounted, and sliced in ice-cold NMDG ACSF on a Leica VT1200s vibratome or a Vibratome 3000 vibratome. Slices were incubated in NMDG solution at 34 °C for 12–15 min before being transferred to room temperature HEPES holding ACSF of the following composition for 45–60 min before experimentation began: 92 mM NaCl, 2.5 mM KCl, 1.25 mM NaH$_2$PO$_4$, 30 mM NaHCO$_3$, 20 mM HEPES, 25 mM glucose, 2 mM thiourea, 5 mM Na-ascorbate, 3 mM Na-pyruvate, 2 mM CaCl$_2$·4H$_2$O and 2 mM MgSO$_4$·7H$_2$O, 12 mM NAC, titrated pH to 7.3–7.4 with concentrated hydrochloric acid, 300–310 mOsm. All solutions were continuously perfused with 5% CO$_2$/95% O$_2$ for 20 min before use.

For VSDI experiments, slices (400 μm thickness) were prepared from n=13 Long–Evans rats (100–150 g). Before the procedure, the rats were anesthetized with isoflurane (Isofane, Vericore), before being decapitated. Brains were extracted from the skull and placed into an oxygenated (95% O$_2$–5% CO$_2$) ice-cold solution of ACSF, made with the following (mM): 124 NaCl, 5 KCl, 1.25 NaH2PO4, 2 MgSO4, 2 CaCl2, 10 glucose, 22 NaHCO3. The brains were sectioned at an oblique horizontal plane (front tilted ~5 degrees downwards).

Slices were then moved to a fine-mesh membrane filter (Omni pore membrane filter, JHWP01300, Millipore) held in place by a thin Plexiglas ring (11 mm inner diameter; 15 mm outer diameter; 1–2 mm thickness) and kept in a moist interface chamber, containing previously used ACSF and continuously supplied with a mixture of 95% $O_2$ and 5% $CO_2$ gas. Additionally, the slices were kept moist from gas being led through ACSF before entering the chamber. The ACSF was kept at 32 °C. Slices were allowed to rest for at least 1 hr before use, one by one in the recording chamber superfused with ACSF.

## Cell identification and electrophysiological recording

Individual slices were transferred to a submersion chamber continuously superfused with bath ACSF of the following composition: 119 mM NaCl, 2.5 mM KCl, 1.25 mM $NaH_2PO_4$, 24 mM $NaHCO_3$, 12.5 mM glucose, 2 mM $CaCl_2 \cdot 4H_2O$ and 2 mM $MgSO_4 \cdot 7H_2O$, titrated pH to 7.3–7.4 with concentrated hydrochloric acid, 300–310 mOsm, held at 32 °C, and perfused with 5% $CO_2$/95% $O_2$ for 20 mins before use. Neurons were visualized with a digital camera (Hammamatsu ORCA-Flash4.0 V3 C13440) and imaged under an upright microscope (Sutter Instruments) using 10 X (0.3 NA, Olympus) and 40 X (0.8 NA, Zeiss) objective lenses with transmitted infrared light or epifluorescence in various wavelengths.

CLA neurons were identified in acute slices by one of several methods. First, in the majority of experiments, neurons were patched within the subregion of retrogradely labeled somas following CTB injection in the RSP (*Figure 1*, *Figure 1—figure supplement 3*). Additionally, in most experiments, we also used fluorescently-labeled corticoclaustral axons (*Figure 3*) from two different sources (*Figure 4*) to further identify the CLA. In a small subset of experiments in Nkx2.1-Cre;Ai9 animals, we were also able to visualize a tdTomato-labeled dense plexus of fibers in the CLA that matches with previous identifications of the CLA relying on a dense plexus of parvalbumin-positive fibers (*Marriott et al., 2021*; *Kim et al., 2016*; *Druga et al., 1993*; *Real et al., 2003*; *Real et al., 2006*; *Du et al., 2008*).

Borosilicate glass pipettes (4–8 MΩ, 1–3 µm tip outer diameter) were pulled using a Narishige PC-10 two-step puller with steps at 65.1°C and 44.2°C and filled with an intracellular solution for electrophysiological recordings of one of the following compositions: (1) 128 mM K-Gluconate, 10 mM HEPES, 4 mM NaCl, 5 mM Mg-ATP, 0.3 mM $Li_2$-GTP, 2 mM $CaCl_2$, 8.054 mM biocytin, pH 7.2, 285–290 mOsm. (2) 110 mM Gluconic acid, 40 mM HEPES, 5 mM $MgCl_2$, 0.2 mM EGTA, 2 mM ATP, 0.3 mM GTP, 5 mM lidocaine, 8.054 mM biocytin, pH 7.2 with CsOH, 285–290 mOsm.

Whole-cell patch-clamp recordings were made from single neurons using a Multi-Clamp 700B amplifier (Molecular Devices) in current-clamp mode and controlled with custom protocols in PackIO (*Watson et al., 2016*). Briefly, neurons were approached in voltage-clamp (0 mV) with intracellular solution back-filled pipettes under positive pipette pressure and ×40 magnification. Negative pressure was applied once a small dimple in the membrane could be seen and was held (–60 mV) until >1 GΩ seal had formed, after which the seal was broken and recording began. Recordings were low-pass filtered at 10 kHz and digitized at 10 or 20 kHz. Results were not corrected for the –14 mV liquid junction potential during current-clamp experiments and the –0.69 mV junction potential in voltage-clamp experiments. The chloride reversal potential in each case was –72 mV and –65 mV, respectively.

To be included for further analysis, patched neurons needed to pass several quality-control criteria during the recording of intrinsic profiles. These included $R_{access}$ <35 MΩ or <20% of $R_{input}$, relative action potential amplitude at rheobase >50 mV and an absolute amplitude above 0 mV, $I_{hold}$ > –30 pA, absolute drift from baseline (measured from the beginning of the recording)<10 mV, and a resting membrane potential < –50 mV (see *Figure 1—figure supplement 3*).

For pharmacological control experiments, neurons were patched in voltage clamp mode, and recordings were made at –70 mV and 0 mV before application of TTX (1 µM) and 4AP (1 mM) to isolate monosynaptic currents. Further recordings additionally included DNQX/APV (50 µM) to block excitatory currents.

In most cases, only one CLA neuron was patched per slice to prevent ambiguity during morphological reconstruction. Once recordings were complete, neurons were allowed to fill with biocytin for up to 30 min, after which the pipette was withdrawn from the tissue, and slices were transferred to 4% paraformaldehyde (PFA).

## Photostimulation of chrimsonR and chronos

In experiments where opsin-expressing viruses were injected into either the cortex or CLA, several different optogenetic photostimulation protocols were used. Briefly, 470/595 nm LEDs were used to deliver light pulses (4ms or 500ms) through a 40 X objective lens. LED power on the sample was titrated to the minimum power required to elicit a response in each cell. 470 nm LED power under the objective lens ranged between 0.069 mW and 3.99 mW and was typically 0.6 mW. 595 nm LED power under the objective lens ranged between 0.61 mW and 4.4 mW and was typically 1.22 mW. Except for dual-color sequential stimulation, all light pulses were separated by 10 s to allow sufficient time for opsins to resensitize.

For dual-color sequential photostimulation, contributions of each cortical presynaptic input axon expressing either Chrimson or Chronos were assessed separately by photostimulation with 470 or 595 nm light pulses (4ms). To disambiguate 470 nm-evoked Chrimson responses from 470 nm-evoked Chronos responses, 595 nm light was pulsed for 500ms followed immediately by a brief 4ms 470 nm pulse to desensitize Chrimson opsins expressed in presynaptic terminals before Chronos stimulation. All photostimulation experiments were repeated ten times and averaged.

For experiments in which FLEX-Chrimson was expressed directly in CLA neurons via retro-Cre injection into RSP, non-expressing CLA neurons were patched and stimulated using 595 nm light (4ms) at 0.1 Hz. 595 nm light was typically set at 1.22 mW power on sample. The same protocol was used during the voltage-clamp recording of cortical neurons in response to CLA axon stimulation.

## Morphological recovery

Patched tissue was fixed in 4% PFA for 2 hr or overnight as described above. Sections were then removed from PFA and washed 3x5 min in 0.01 M phosphate-buffered saline (PBS). Sections were then transferred to 0.01 M PBS and 0.25% TritonX (PBST) and allowed to incubate in streptavidin for at least 3 days (1:500 Streptavidin, Alexa Fluor 488/647 conjugate, Thermo Fisher S11223/S21374). The tissue was then washed 3x5 min in 0.01 M PBS, mounted, coverslipped, and imaged as described below.

## Perfusion and tissue sectioning

Mice were deeply anesthetized with 5% isoflurane before receiving an overdose of pentobarbital via intraperitoneal injection. Mice were then transcardially perfused with 0.01 M PBS, followed by 4% PFA. The brain was then extracted and allowed to fix in 4% PFA overnight. Brains were then moved to 0.01 M PBS and mounted for sectioning on a Leica VT1000s vibratome. Slices were sectioned coronally to 50 μm or 100 μm thickness and placed in 0.01 M PBS before immunohistochemistry and mounting or stored in tissue freezing solution (45% 0.01 M PBS, 30% ethylene glycol, 25% glycerol) at –20 °C for up to 3 years.

## Immunohistochemistry and imaging

Mice were perfused and sections were collected as above. Sections were first washed 3x5 min 0.01 M PBS before permeabilization in 0.5% PBST for 2x10 min. Sections were then blocked for 90 min in PBST and 5% normal goat or donkey serum at room temperature, after which they were incubated in primary antibody (mouse anti-MBP 1:500, Merck NE1019, RRID:AB_604550; rat anti-MBP 1:500, Abcam ab7349, RRID:AB_305869; rabbit anti-FLAG 1:500, Cell Signaling Technology 14793, RRID:AB_2572291; mouse anti-FLAG 1:500, Cell Signaling Technology 8146, RRID:AB_10950495; rabbit anti-GFAP 1:500, Merck G9269, RRID:AB_477035; rabbit anti-PV 1:400–500, Swant PV27a, RRID:AB_2631173; chicken anti-GFP 1:2500, AVES GFP-1020, RRID:AB_10000240) for at least 48 hr at 4 °C. The slices were then washed 3x5 min in 0.5% PBST followed by incubation in secondary antibodies (goat anti-mouse Alexa Fluor 405 1:500, Invitrogen A31553, RRID:AB_221604; goat anti-rabbit Alexa Fluor 488 1:500, Invitrogen A11034, RRID:AB_2576217; donkey anti-rabbit Alexa Fluor 594 1:500, Jackson ImmunoResearch 711-585-152, RRID:AB_2340621; Donkey anti-chicken Alexa Fluor 488 1:500, Jackson ImmunoResearch 703-545-155, RRID:AB_2340375) for 3 hr. Finally, the tissue was washed in 0.01 M PBS for 3x5 min, then mounted and coverslipped.

For tissue that was to be stained for GABA, mice were perfused and sections were collected as above, but 0.25% glutaraldehyde was added to 4% PFA for fixation. Sections were first washed 3x5 min 0.01 M PBS before permeabilization in 0.2% PBST 30 min. Sections were then blocked for

3 hr in PBST and 5% normal goat or donkey serum at room temperature, after which they were incubated in primary antibodies (rabbit anti-GABA 1:500, Sigma-Aldrich Cat# A2052, RRID:AB_477652) for at least 72 hr at 4 °C. The slices were then washed 3x5 min in 0.2% PBST followed by incubation in secondary antibodies (For Cre-injected mice, goat anti-rabbit Alexa Fluor 488 1:200, Thermo Fisher Scientific Cat# A-11034, RRID:AB_2576217; For CTB-injected mice, donkey anti-rabbit Alexa Fluor 488 1:200, Thermo Fisher Scientific Cat# A-21206, RRID:AB_2535792) for at least 48 hr at 4 °C. Finally, the slices were washed in 0.01 M PBS for 3x5 min, then mounted and coverslipped.

Once dry, whole-slice and CLA images were taken at ×4 and ×10 magnification (UPlanSApo, 0.16 and 0.4 NA) on an Olympus FV3000 laser scanning confocal microscope. For recovered morphologies, images were taken on the above microscope or a Zeiss LSM710 confocal laser scanning microscope at ×20 magnification and tiled across the z-axis, or on a custom 2 p microscope at ×16 magnification (Coherent Vision-S laser, Bruker 2PPlus microscope, Nikon 16X0.8 NA objective). Images of post-hoc histology from calcium imaging animals were taken on a Ziess LSM710 at ×10 magnification tiled across the z-axis, or a Leica epifluorescence microscope (1.6 X). Slices and morphologies were not corrected for tissue shrinkage as a result of fixation. Histological images from animals in the behavioral experiments were taken using a Zeiss Axioscan Z1 automated slide scanner with a 10 x objective.

GFAP expression was quantified using FIJI (ImageJ). First, ovoid ROIs were drawn surrounding the left and right ventral claustrum of each section. The ROI was drawn by hand using only the MBP image and without reference to the GFAP channel. Next, an ROI for the entire slice was drawn using the default algorithm of the thresholding tool. The mean fluorescence intensity was then recorded for all ROIs. The GFAP fluorescence was then averaged across the two claustrum ROIs on each slice and normalized against the fluorescence of the entire slice.

## Voltage-sensitive dye imaging

Slices were stained for 3 min with VSD RH-795 (R649, Invitrogen, 0.5% in ACSF) and imaged in a recording chamber positioned beneath a fluorescence microscope (Axio Examiner, Zeiss). The slices were excited with 535±25 nm light (bandpass), reflected by a dichroic mirror (half reflectance wavelength of 580 nm), and epifluorescence was detected using a long-wavelength pass filter (50% transmittance at 590 nm) with a CMOS camera (MiCAM Ultima, BrainVision, Japan; 100_100 pixel array). An electronically controlled shutter built into the light source (HL-151, Brain Vision) was set to open for 500ms before the optical recording was triggered, as a way of avoiding mechanical disturbance caused by the shutter system and rapid bleaching of the dye.

The optical baseline was allowed to stabilize for 50ms before the delivery of any stimulus. 512 frames at a rate of 1.0ms/frame were acquired in all experiments. Color-coded optical signals were superimposed on the brightfield image to represent the spread of neural activity. The fraction of the optical signal that exceeded the baseline noise was displayed as a heatmap. Baseline noise was reduced by averaging eight identical recordings acquired with a 3 s interval directly in the frame memory. The optical signals were analyzed using BrainVision analysis software. Changes in membrane potential were evaluated in a region of interest (ROI) as the fraction of change in fluorescence (dF/Fmax%), where Fmax equals the highest fluorescence value during the baseline condition of each stimulation. Based on visual inspection of the optical signal, the ROI was chosen as the region where the signal first entered the CLA. The stimulation electrode was a tungsten bipolar electrode with a tip separation of 150 µm.

A total of 28 recordings were used in these experiments. Stimulations were either elicited at a rostral position of the CL with the signal propagating in the caudal direction or at a caudal position with the signal propagating in the rostral direction. In cases where a single pulse did not elicit measurable activation in the CLA, four or five repetitive stimulations (0.1–0.3 mA, 300 µs, 40 Hz) were used. At least five stimulation cycles were repeated for all experiments to assess if activation had occurred. Latency was measured from the beginning of the stimulus artifact to the onset of the response in the CLA. In some experiments, the recording ACSF for VSD imaging contained a low dose of 25 µM DNQX and 50 µM APV. Electrodes and parameters, as well as analyses, were based on prior studies. Following the VSD experiments, slices were postfixed in 4% PFA for up to 1 week, before being transferred to a PBS solution with 30% sucrose. Then after at least 10 hr they were cut at 40–50 µm thickness using a freezing microtome. Sections were then mounted and Nissl-stained with Cresyl Violet

before coverslipping with Entellan. Images of the sections were combined with the optical imaging data to identify the ROIs from the recordings.

## Optical system for in vitro visualization/photostimulation

The optical system used for in vitro visualization and photostimulation combined blue (Thorlabs M470L4), orange (Thorlabs M595L3), and far-red (Thorlabs M625L3) LED paths. Briefly, orange and far-red LED paths were combined via a 50/50 beamsplitter (Thorlabs BSW10R), then passed through a blue/red combining dichroic mirror (Thorlabs DMLP505R). Light was then passed down onto the sample through either an RGB dichroic mirror (Laser2000 FF409/493/573/652-Di02−25x36) for epifluorescence visualization or a 'cold' mirror (Thorlabs FM03R) for photostimulation. Tissue was visualized via 850 nm light transmitted through a condenser mounted beneath the slice chamber (Thorlabs M850L3). Incident and reflected light passed through excitation (Semrock FF01-378/474/554/635-25) and emission (Semrock FF01-432/515/595/730-25) filters while in RGB visualization mode.

### Optical system for in vivo visualization of CLA axons

All two-photon imaging was performed using a Bruker Ultima 2*P*+two-photon microscope controlled by Prairie View software, and a femtosecond-pulsed, dispersion-corrected laser (Chameleon, Coherent). Imaging was performed using a Nikon 16X0.8 NA water immersion lens. The lens was insulated from external light using a custom 3D printed cone connected to a flexible rubber sleeve. A wavelength of 920 nm and 50 mW power on the sample was used for visualizing GCaMP7b. An imaging rate of 30 Hz and a 512x512 pixel square field of view (FOV) were used for all recordings. FOVs were selected across the right side of the cranial window. The approximate coordinates of the center of the FOV relative to bregma ranged from: AP –1.2 to +1.3; ML +0.3 to+1.5; DV –0.03 to –0.3.

## In vivo sensory stimulation

Once mice had completely recovered from surgery, and after allowing sufficient time for viral expression (>3 weeks), mice were assessed for GCaMP7b labeled axons in the cortex. Animals were first acclimated to head fixation under the microscope. Next, GCaMP7b expression levels were assessed by eye. Animals in which no GCaMP7b-labeled axons could be found in the cranial window were excluded from future experiments. Animals with GCaMP7b-labeled axons in the cortex were then used for multimodal stimulation experiments.

Sensory stimuli were delivered using a data acquisition card (National Instruments) and PackIO software. Briefly, custom MATLAB (MathWorks) code was used to generate voltage traces. These traces were then used by PackIO to output timed voltage from the data acquisition card to an LED (Thorlabs MNWHL4), a piezoelectric whisker stimulator (Physik Instrumente), and a speaker (Dell). Stimuli lasted 500ms. The light stimulus consisted of a flash of white light (~5.5 mW emitted ~20 cm to the right of the mouse), the auditory stimulus of an amplitude and frequency modulated complex tone with a 5 kHz carrier frequency, and the tactile stimulus of a paddle oscillating at , 20 Hz within the mouse's right whiskers.

During each experiment, mice were first head-fixed under the microscope. Imaging was performed in an enclosed hood to minimize visual stimuli, and white noise was used to obscure extraneous sounds. The surface of the cranial window was leveled relative to the imaging plane using a tip-tilt stage (Thorlabs). During each imaging session, FOVs with visible axon expression were selected manually. In the unimodal-only cohort (*Figure 6—figure supplement 1*), mice were presented with 60 randomly interleaved stimulus presentations separated by randomly generated 8–11 s intertrial intervals. These 60 stimuli were randomly drawn from 4 trial types: sound, light, whisker, and blank. In the uni- and multimodal cohort, mice were presented with 120 stimuli randomly drawn from 8 trial types: sound alone; light alone; whisker alone; sound and light; sound and whisker; light and whisker; sound, light, and whisker; and blank. During blank trials, no stimuli were delivered. The order and the precise number of each trial type were randomly generated each day. After all stimuli were delivered, a new FOV was then selected, and the sensory stimulation was repeated. Imaging FOVs were selected based on visible axon expression and were drawn from across the extent of the cranial window. Care was taken to avoid recording from the same axon twice on a given day. However, as axons were only visible when active, and given the contorted and branched shape, separate ROIs may have included the same axons.

The first round of unimodal data collection involved five mice (*Figure 6—figure supplement 1* M1–4), of which one was excluded due to unrecoverable histology (not shown). The second round of data collection involved seven mice (*Figure 6—figure supplement 1* M4–9), of which one was excluded due to unrecoverable histology (not shown), and two were excluded due to off-target expression in cortex (*Figure 6—figure supplement 1* M8–9). Two animals were used in both data sets (*Figure 6—figure supplement 1* M4), of which one was excluded due to unrecoverable histology (not shown).

## Behavioral testing

The behavioral effects of CLA silencing were investigated in two experiments. In the first experiment, animals with sham and active CLA silencing were compared using 24/7 home cage activity monitoring. In the second experiment, animals with sham and active CLA silencing were compared using two reward-motivated tasks.

### Home cage activity monitoring

Animals aged p41–p53 were single housed in activity tracking cages. After a two-week acclimation period, animals were returned to standard cages for 5–8 days to receive bilateral intracranial injections of either retro-Cre in RSP and FLEX-TetTox in CLA (TetTox; n=6) or equivalent volumes of PBS (Sham; n=6). Mice were then returned to the activity tracking cages for a further six weeks of monitoring. Finally, mice were tested on a series of ethologically motivated tests for anxiety-like behavior (elevated plus maze, aversive open field, and light-dark box). Given the sex-specific effects of social isolation, only male mice were used in this experiment (*Oliver et al., 2020*).

### Activity tracking cages

During the study, mice were housed in Digital Ventilated Cages (DVC; Techniplast). DVCs resembled standard individually ventilated cages, but cage racks included capacitive sensors which permitted continuous, passive, and non-disruptive behavioral monitoring. The sensor consisted of 12 capacitive plates arrayed underneath each cage. Readouts from each of the 12 sensors were recorded every 250ms and enabled tracking of both general locomotor activity as well as distance traveled. Further details about the DVC system and data analysis can be found below as well as in *Pernold et al., 2021* and *Iannello, 2019 Iannello, 2019*; *Pernold et al., 2021*.

### Elevated plus maze

The apparatus consisted of four 34x7 cm arms joined together at a central square. Two opposing arms were enclosed by 20 cm walls while the other arms were left exposed. The apparatus was raised 50 cm above the ground. Mice were placed at the end of one of the enclosed arms and allowed to explore freely for 5 min.

### Aversive open field

The apparatus consisted of a brightly illuminated 60 cm diameter circular arena. Mice were placed into the arena and allowed to explore for 10 min. The apparatus was divided into three zones: center, intermediate, and outer. The center zone was defined as a circle in the middle of the arena with a 5 cm radius. The outer zone was defined as a 5 cm band running around the extreme outside of the arena. Finally, the intermediate zone consisted of the area between the outer and center zones. The mouse's location within the arena was tracked throughout the test to measure its preference for the center, intermediate, and outer zones.

### Light dark box

The apparatus consisted of a 27x27 cm box split into two chambers connected through a small doorway. One chamber was open and brightly lit while the other was covered and dark. Mice were placed initially into the dark box and allowed to explore for 5 min. The time spent in the light box was recorded to determine animals' preference for the light or dark areas.

## Reward-motivated behavioral tasks

*Figure 7D* provides a schematic outline of the experimental structure. Briefly, animals aged p44–p48 received bilateral intracranial injections of either retro-iCre-mCherry in RSP and FLEX-TetTox in CLA (TetTox; n=16) or equivalent volumes of PBS (Sham; n=12). Animals were then given four weeks for recovery and viral expression. Mice were then water-restricted and trained on a multimodal conditioning task before a 1-week break from water restriction. Next, mice were again water-restricted for 2 days before being trained on a reversal learning task. Finally, mice were sacrificed, and brains were extracted for post-hoc histology. Both groups contained equal numbers of male and female mice.

Water restriction was used to provide task motivation. Mice were weighed twice daily during water restriction (before and after training) to ensure that their weight remained above 85% of baseline. On the first and second days of water regulation, mice were given access to their water bottle for 1 hr each day. Training commenced on the third day, after which most mice received all their water during the behavioral task. Mice who did not receive sufficient water during their training sessions or whose weight dropped below 85% of baseline were given additional water outside of training to maintain their body weight.

## Behavioral training setup

Training took place in 12x12 cm plastic boxes with nine nose-poke ports on the back wall arranged in a diamond shape (see https://github.com/pyControl/hardware/tree/master/Behaviour_box_small; *Akam, 2021*). Removable plastic panels could be attached to the back wall to cover various combinations of nose poke ports. Each port contained an infrared beam to detect nose pokes. Solenoid valves could be used to deliver calibrated water rewards to each port. Each box was housed in a separate sound and light attenuating chamber. A speaker located above the nose poke ports was used to deliver auditory stimuli. Each port could be illuminated individually, and the entire chamber could be illuminated with an LED located above each box. Operant boxes were controlled and programmed using pyControl (*Akam et al., 2022*).

## Multimodal conditioning

Water-restricted mice were trained to associate uni- and multimodal sensory stimuli with water rewards. Mice were trained in two sessions per day for 11 days with only one session on the last day. Each session included 30 trials and lasted for ~50 min. Trials were separated by a random 1–2 min inter-trial interval. During each trial, mice were presented with either an auditory stimulus (A), a visual stimulus (V), or both stimuli at the same time (AV). Each stimulus was presented for 10 s. The A consisted of a stepped sine wave rising from 5 to 15 kHz. The frequency was increased in 20 steps, and the steps were cycled at 50 Hz, meaning that each step was played for 20ms. The V consisted of turning on the house lights in the behavior box. The AV consisted of the A and V presented simultaneously. Each stimulus was associated with reward availability at a different port, with the A stimulus indicating reward availability at the left port, the V at the right port, and the AV at the center port. Rewards were delivered following the first poke to the correct port following stimulus onset, regardless of whether the subject initially poked an incorrect port. Rewards available in a given port were not cumulative, that is subjects did not receive two rewards if they failed to collect the reward before the next presentation of the same stimulus type. Each water reward was 15 uμl. This was decreased to 12 μl for mice whose weight remained at or above 100% of baseline for two consecutive sessions.

## Reversal learning

Water-restricted mice learned to flexibly track probabilistic associations between action and reward across reversals in reward probability. Mice were trained in 2 sessions per day for 10 days. Each session lasted 45 min, during which time mice could complete trials to earn rewards.

Training took place in the same boxes as the multimodal conditioning. All poke ports used in the previous paradigm were covered, and an upside-down triangle of poke ports was uncovered. At the start of the training session, the house light was illuminated and remained illuminated throughout training. Trials were separated by a 1 s inter-trial interval. At the end of the inter-trial interval, the two top poke ports were illuminated. Mice could then make a choice by poking either the left- or right-hand poke port. A click was played after poking either port. After making a choice, the top two port

lights were turned off and the bottom port was illuminated. Mice could then poke the bottom port for a chance to receive a water reward. Reward volumes were gradually decreased over the course of training to maintain motivation. At the start of the first session, the choice ports were randomly assigned as either the good or bad port. Poking the good port led to an 85% chance of receiving water from the reward port, while poking the bad port led to a 15% chance of receiving a reward. Each mouse's performance was tracked throughout the session using an exponential moving average (tau = 12). A reversal in reward probabilities was triggered 5–10 trials after mice crossed a threshold of >80% correct. The reward probabilities at the end of a session were used at the start of the next session. The number of trials and blocks completed and the number of rewards obtained in each session were therefore dependent on each mouse's individual performance.

## Data analysis and availability

All analyses were performed with custom routines using Python 3.7.9 or open source packages unless otherwise stated. Processed data and the functions used to generate the figure panels in this study are available at https://github.com/AMShelton/claustrum-integration (copy archived at *Shelton, 2025*).

### Electrophysiological analysis

Intrinsic electrophysiological recordings taken in current-clamp mode were passed through a series of automated quality controls (see *Figure 1—figure supplement 3*) before features were calculated and stored for later cell-typing analysis. These included: access resistance less than 35 MΩ or less than 20% of measured input resistance, membrane voltage ($V_m$) less than –50 mV, $V_m$ drift less than 10 mV, threshold action potential amplitude greater than 50 mV from spike onset, and an absolute holding current of less than 30 pA. Measured access resistance across cell types was found to be similar (see *Supplementary file 1*). All extracted feature data is publicly available (see Data Availability Statement).

Neurons that passed quality controls were first sorted manually based on their intrinsic profiles at threshold and 2 x threshold current injection. Excitatory and inhibitory neurons were segregated and then independently classified into further subgroups. Automated classification involved a preprocessing step in which the electrophysiological dataset was standardized for every feature and cell. Principal component analysis (PCA, 53 features) was then used to reduce the dimensionality of the standardized dataset, producing a neurons x components matrix, the first three components of which accounted for greater than 85% of the explained variance in the dataset. All components were used in uniform manifold approximation and projection (UMAP), the data from which was plotted and clustered using k-means clustering. Clusters were compared via silhouette analysis, and the average silhouette score across samples was used as an indicator of how well unsupervised methods had identified which value of k best represented electrophysiological groups.

Due to poor separation among subclasses of excitatory neurons using the above method, we compared manually sorted groups of excitatory neurons using a select set of electrophysiological features. Feature comparisons for which the correlation between the features was high ($r$>0.7; e.g. spike rise time and spike rise rate) were ignored for this analysis. To determine if a neuron was excitatory or inhibitory, we used the presence of spines in *post-hoc* histology, a biphasic AP waveform, 200 MΩ≤$R_{input}$ ≤ 500 MΩ, AP half-width ≥1.0ms, or a combination of the above. E1 and E2 neurons were further segregated by the presence or absence of an ADP and the monophasic or biphasic pattern of their AP waveform at suprathreshold current injections (*Figure 1—figure supplement 4*). Inhibitory neurons were segregated by unsupervised clustering only.

For in vitro optogenetic mapping experiments, ten trials for each cell were taken and averaged. Response magnitudes relative to baseline were calculated as the difference in the integral of the post-stimulus (30ms after stimulus offset) and pre-stimulus (30ms before stimulus onset) periods. Significant responses in current-clamp and voltage-clamp modes were taken as those exceeding three and five standard deviations from the average baseline period, respectively, and validated manually and by Mann-Whitney U tests that were corrected for multiple comparisons via Benjamini-Hochberg false discovery rate analysis with an alpha of 10%. Latencies for significant and non-significant responses were found manually. Neurons with evoked spike latencies shorter than 3ms were taken to be directly expressing opsin and were removed from analysis (*Figure 2—figure supplement 1*).

To determine the expected probability of integration against what was observed, we found the probability of the linear combination of response probabilities to each cortex, assuming the independence of each input. The expected probability of integration was defined as:

$$p\left(expected\right) = p\left(input1 + input\left(1 + 2\right)\right) * p\left(input2 + input\left(1 + 2\right)\right)$$

Where input1 and input2 are the occurrences of a post-synaptic response to a given cortex (e.g. cortex1 and cortex2) and input(1+2) is the occurrence of responses to both tested cortices in the same cell.

## Morphological reconstruction analysis

Images of filled neurons were processed using ImageJ (v1.8.0_172), then uploaded to the software Neurolucida 360 (MBF) and used as a template for semi-automated, user-guided reconstruction in three dimensions. Neurolucida Explorer (MBF Bioscience) was used to extract a range of dendritic, somatic, and axonal properties from neuronal reconstructions. Dendritic spines were counted only on neurons with fills of sufficient quality. Spine quantification proceeded by manually counting the number of spines within a 100 μm section along three primary and three secondary dendrites, the averaging. Spiny and sparsely spiny neurons were categorized together as 'spiny.' All cell data was compiled into a morphological dataset that is publicly available (see Data Availability Statement).

## 2-photon calcium imaging analysis

Calcium imaging data were preprocessed using Suite2P to remove motion artifacts (*Pachitariu et al., 2016*). For the unimodal cohort, axonal ROIs were selected by hand using ImageJ. For the uni- and multimodal cohort, axonal ROIs were automatically selected using Suite2P. Automatically generated ROIs were then curated manually. ROIs were selected based on their morphology and activity traces. We computed ΔF/F for each axon using the equation:

$$\Delta F/F = (F - F)/F$$

where $F$ represents the mean of $F$ across time through the entire session. For axonal ROIs selected by suite2P, $F$ was first corrected for neuropil fluorescence by subtracting 0.7*FNeu.

After calcium traces were exported from Suite2P, all analyses were carried out using custom MATLAB code. Calcium traces were plotted using the Gramm software package (*Morel, 2018*).

Extracted calcium signals were then analyzed to identify axon segments that significantly responded to one or more sensory modalities. First, the calcium signal from 2 s before to 6 s after stimulus onset was averaged for all presentations of a given trial type (i.e. whisker alone, whisker and sound) for each axon segment in each FOV. Significantly responsive axon segments were identified by using a non-parametric Mann-Whitney U test to compare the signal in the second before and after stimulus onset. Multiple comparisons correction was performed using the Benjamini-Hochberg false discovery rate analysis with an alpha of 1%.

Responsive axons were classified as either uni- or multisensory based on the trial types to which they responded. Unisensory axons were those which responded to only one modality and whose response was not modulated by other modalities (ex. an axon which responded to all trial types that included the sound stimulus and none which did not include the sound stimulus). Multisensory axons were those which either (a) responded to multiple sensory modalities (including those which only responded to multimodal trial types) or (b) whose response to a unimodal trial type was modulated by the addition of other modalities (ex. an axon which responded to the unimodal sound stimulus but did not respond to multimodal trial types that included the sound stimulus).

To understand the trial-to-trial diversity of axonal responses, we calculated the AUC for the dF/F in the second after stimulus onset for each axon. To measure the reliability of responses, we computed the response probability for each axon number of stimulus presentations evoking a response/total number of stimulus presentations. Stimulus presentations were deemed to evoke a response if the AUC was >1 standard deviation above the mean AUC for all axons, baselined on a trial-by-trial basis to the second before stimulus onset. A non-parametric Kruskal Wallis test was used to assess the impact of trial type and/or session on axonal response probability and magnitude.

## Confocal image analysis

All images used for quantitative analysis in this study were imaged on a confocal microscope at ×10 magnification (see Immunohistochemistry and imaging section above for details). Cell counts and cell coordinates were collected and analyzed using ImageJ and custom JavaScript macros. Inter-cell distances were calculated as the smallest Euclidean distance between cell somas. Comparisons between cell counts from injection sites were done using Mann-Whitney U tests, corrected for multiple comparisons by Bonferroni correction.

Contours generated from confocal images of CTB +neurons in the CLA (n=3 mice) were made via morphological snakes (*Caselles et al., 1995*; *Kass et al., 1988*) of average images. Briefly, confocal images of the CLA taken from 50-μm-thick sections were thresholded using Otsu's method (*Otsu, 1979*). A binary erosion algorithm (*Gonzalez and Woods, 2008*) was applied to thresholded images to remove noise from small, punctate autofluorescence above threshold in each image. Processed images were then multiplied by their original counterparts to create denoised, native fluorescence intensity images of CTB +CLA neurons. Images of the same slice across mice were grouped based on the Paxinos & Franklin Brain Atlas and the Allen Brain Atlas (*Franklin and Paxinos, 2008*; *Wang et al., 2020*). Each image in a group representing a single coronal plane was normalized and aligned to the center of mass (COM) of fluorescence before being averaged into a single image. After the generation of these average images for each AP plane, the border of the CLA as defined by CTB +neurons was found by initializing an ellipse about the COM of CLA fluorescence to act as a boundary for morphological snake active contours. These contours evolve in time and are pulled toward object boundaries until the energy functions reach their minimum. Area for each contour was calculated as the integral for the closed contour path.

Confocal images of cortical axons innervating the CLA were prepared as above and COM-aligned to the CTB signal in the CLA across mice for a given cortical injection (n=3 mice/injection site; analyzed image in *Figure 3f*, *Figure 3—figure supplements 1 and 2* from ~1.00 mm bregma). Axonal fluorescence from each image was normalized and averaged to 15 μm x 15 μm bins and displayed as a heatmap.

To determine the amount of dorsal/core/ventral fluorescence within the CLA of each injection experiment, the CLA contour at the AP position of the analyzed image was used as a mask for the core. Dorsal and ventral masks were taken as the regions in the image above and below the core, including medial and lateral regions above and below ½ the core height. Image masks were multiplied to each processed and normalized image within an injection experiment set, and fluorescence from that region was averaged pixel-wise. Regional fluorescence was then averaged across mice to obtain a comparison of dorsal, core, and ventral axon fluorescence in the CLA from each cortical area. Values for each region were compared using independent t-test. Multiple comparisons correction was performed using the Benjamini-Hochberg false discovery rate analysis with an alpha of 10%.

## Home cage activity tracking

The home cage monitoring study was divided into baseline and expression periods. To allow mice to acclimatize to their new environment, data for the baseline period was collected starting one week after mice were placed in the activity monitoring cages until they were removed for surgery. To balance the baseline period and allow adequate time for viral expression, data for the expression period was collected for 1 week starting 3 weeks after the last mouse in each group received their intracranial injections.

## Home cage activity

Activity tracking data from the DVC system were preprocessed using the DVC Analytics platform (Techniplast) to extract locomotor activity data. Briefly, readouts from the capacitive plates are affected by the local electromagnetic environment. The presence of water-rich bodies — such as a mouse — near the sensor alters its readout. Alterations to the capacitive readings across the 12 sensors can be used to identify the location of a mouse within the cage. By tracking the change in the location of the mouse across readings, the system can track the distance traveled by the mouse within a given time window.

While the distance traveled metric reveals an important aspect of home cage behavior, it is insufficient to distinguish between a truly immobile mouse (ex. during sleep) and a mouse which is stationary

but active (ex. grooming, nest building). To provide a more detailed picture of activity, the DVC platform also calculates an Animal Activity Index. Activity is detected by computing the change in electrode capacitance between adjacent readings. An electrode is considered activated if the difference between the readings is greater than a threshold chosen by the software to separate activity from noise. This measurement is therefore insensitive to long-term changes in capacitance (ex. due to the moisture level of the bedding) while providing a short-term readout of local activity. The density of activation across the electrode array over time can then be used as a general indicator of the animals' activity level. The so-called Animal Activity Index calculated by the DVC analytics platform is expressed in arbitrary units.

## Circadian parameters

Python, the pyActigraphy package, and the DVC analytics platform were used to compute circadian metrics from the DVC data. While the pyActigraphy toolbox was written to analyze data collected using devices such as smart watches, it was adapted here to import data produced by the DVC analytics platform. The pyActigraphy toolbox was then used to analyze animals' mean inter-daily stability (ISm) and mean intra-daily variability (IVm). The DVC data were further analyzed using custom Python code to calculate the relative amplitude (RA) of animals' daily activity. RA was calculated using the following formula:

$$RA + \frac{AI_D - AI_L}{AI_D + AI_L}$$

where $AI_D$ refers to animals' mean activity index during their dark cycle and $AI_L$ refers to their mean activity in the light cycle. These metrics were calculated using the Animal Activity Index.

Finally, the DVC analytics platform was used to calculate the regularity disruption index (RDI) for all mice at baseline and after claustrum silencing. RDI is a novel biomarker based on sample entropy. It was developed for the DVC system to quantify irregularities in activity patterns. A highly regular pattern of activity would yield an RDI near zero, while highly disrupted activity would yield a high RDI. Notably, RDI values reflect the regularity of activity regardless of its magnitude. RDI was calculated separately for the light and dark cycle to delineate activity patterns in different behavioral states. Details on the above metrics can be found in *Brown et al., 2019*; *Golini et al., 2020*; *Gonçalves et al., 2014*; *Hammad et al., 2021*. Statistical tests were performed using Python and GraphPad Prism 9.

## Activity bouts

Actigraphy data can be used to calculate the number and duration of periods of rest and activity. Activity bouts are usually defined as periods of time when activity never falls below a specified threshold. However, there is variation between studies in the required length and threshold. The parameters of this analysis were selected to maximize the number of bouts detected. Bouts were detected using MATLAB code to identify periods where the Animal Locomotor Index did not fall below a threshold of 0.2 arbitrary units. Bouts had a minimum duration of 1 min with a minimum inter-bout interval of 1 min. Statistical tests were performed using GraphPad Prism 9.

## Anxiety tests

Video recordings of the elevated plus maze and aversive open field tests were analyzed by a blinded observer using Anymaze 7.3 (Stoelting) to extract the mouse's location and movement within the behavioral apparatus. The mouse was deemed to be inside a given region when 85% of its body crossed into the zone. Video recordings of the light-dark box test were analyzed manually by a blinded observer to record time spent in the light zone of the apparatus. Statistical tests were performed using GraphPad Prism 9.

## Multimodal conditioning and reversal learning

Statistical tests were performed using Python and GraphPad Prism 9.

In the multimodal conditioning task, a d-prime (d') was calculated for each stimulus in an effort to compare animals' sensitivity to the different stimuli. To calculate each d', all trials had to be classified

according to one of four outcomes: hit, miss, correct rejection, or false alarm. Classically, a hit is defined as a trial in which both the stimulus and response are present, while a miss is a trial where the stimulus is present, but the response is absent. A correct rejection is a trial in which both the stimulus and response are absent, while a false alarm is a trial in which the stimulus is absent, but the response is present. For this calculation, the response was defined as poking the correct port first, and within 10 s of stimulus onset. Using the auditory d' as an example, a hit would be a trial where a sound was present, and the mouse poked the auditory port first. A miss would be a trial where the auditory stimulus was present, but the mouse poked a non-auditory port first. An auditory false alarm would be a non-auditory trial (i.e. visual or audiovisual) where the mouse poked the auditory port first. Finally, a correct rejection would be a non-auditory trial (i.e. visual or audiovisual) where the mouse did not poke the auditory port first. In this way, each d' included the outcome for every trial of all stimulus types. The redundancy inherent in these outcomes is essential for assessing how animals behave when each stimulus was present and when it was absent.

Once all trials had been assigned an outcome, they were used to calculate a hit rate and a false alarm rate for each stimulus. These were calculated according to the following formulae: False Alarm Rate = ((# False Alarm) / (# False Alarm + # Correct Rejection)) and Hit Rate = ((# Hit) / (# Hit + # Miss)). The hit and false alarm rates could then be combined according to the formula:

$$d' = Z\left(HR\right) - Z\left\{FR\right\}$$

where Z is the normal inverse cumulative distribution function, HR is the hit rate, and FR is the false-alarm rate.

## Acknowledgements

The authors gratefully acknowledge Anna Hoeder-Suabedissen, Huriye Atilgan, Andrew J King, Colin Akerman, Mark Walton, Peter Magill, Zoltan Molnar, Armin Lak, Vladyslav Vyazovskiy, David Bannerman, and Christof Koch for their intellectual insights and discussion in the completion of this project. The authors also thank the Micron Advanced Bioimaging Unit (supported by Wellcome Strategic Awards 091911/B/10/Z and 107457/Z/15/Z), Oxford Behavioural Neurosciences Unit, and the Wolfson Imaging Centre for their support & assistance in this work. Funding for this work comes from the Wellcome Trust to AMP, the European Research Council (ERC) under the European Union's Horizon 2020 research and innovation programme (grant agreement No 852765) to AMP, BBSRC BB/P003796/1 to SJBB, Natural Sciences and Engineering Research Council of Canada (NSERC) to DKO and Clarendon Fund graduate scholarships to AMS and DKO.

## Additional information

### Funding

| Funder | Grant reference number | Author |
|---|---|---|
| Wellcome Trust | 10.35802/204651 | Adam Max Packer |
| European Research Council | 10.3030/852765 | Adam Max Packer |
| Natural Sciences and Engineering Research Council of Canada | | David K Oliver |
| Clarendon Fund | | Andrew M Shelton<br>David K Oliver |
| Biotechnology and Biological Sciences Research Council | BB/P003796/1 | Simon JB Butt |

The funders had no role in study design, data collection and interpretation, or the decision to submit the work for publication. For the purpose of Open Access, the authors have applied a CC BY public copyright license to any Author Accepted Manuscript version arising from this submission.

### Author contributions
Andrew M Shelton, Conceptualization, Resources, Data curation, Software, Formal analysis, Supervision, Validation, Investigation, Visualization, Methodology, Writing – original draft, Project administration, Writing – review and editing; David K Oliver, Data curation, Formal analysis, Validation, Visualization, Writing – original draft, Writing – review and editing; Ivan P Lazarte, Data curation, Formal analysis; Joachim S Grimstvedt, Formal analysis, Visualization, Methodology, Writing – review and editing; Ishaan Kapoor, Jake A Swann, Caitlin A Ashcroft, Simon N Williams, Niall Conway, Selma Tir, Amy Robinson, Data curation; Stuart Peirson, Thomas Akam, Methodology; Clifford G Kentros, Supervision; Menno P Witter, Supervision, Project administration, Writing – review and editing; Simon JB Butt, Conceptualization, Supervision, Funding acquisition, Investigation, Methodology, Writing – original draft, Project administration, Writing – review and editing; Adam Max Packer, Conceptualization, Resources, Supervision, Funding acquisition, Investigation, Methodology, Writing – original draft, Project administration, Writing – review and editing

### Author ORCIDs
Andrew M Shelton ![ORCID] http://orcid.org/0000-0002-5787-4310
David K Oliver ![ORCID] https://orcid.org/0000-0003-1210-8409
Thomas Akam ![ORCID] https://orcid.org/0000-0002-1810-0494
Menno P Witter ![ORCID] https://orcid.org/0000-0003-0285-1637
Simon JB Butt ![ORCID] https://orcid.org/0000-0002-2399-0102
Adam Max Packer ![ORCID] https://orcid.org/0000-0001-5884-794X

### Ethics
Animal procedures were subject to local ethical approval under PPL #PE5B24716 and adhered to the United Kingdom Home Office (Scientific Procedures) Act of 1986. Male and female C57BL/6J or Nkx2.1Cre;Ai9 mice were used in these experiments. Mice were between 3-11 weeks of age when surgery was performed. Long-Evans rats were used in experiments conducted at the Kavli Institute for Systems Neuroscience at the Norwegian University of Science and Technology (NTNU), Trondheim. These experiments were approved by the Federation of European Laboratory Animal Science Association (FELASA) and local authorities at NTNU under FOTS #6269.

Reviewer #1 (Public review): https://doi.org/10.7554/eLife.98002.3.sa1
Reviewer #2 (Public review): https://doi.org/10.7554/eLife.98002.3.sa2
Reviewer #3 (Public review): https://doi.org/10.7554/eLife.98002.3.sa3
Author response https://doi.org/10.7554/eLife.98002.3.sa4

## Additional files

### Supplementary files
Supplementary file 1. Select electrophysiological properties of CLA neurons. All values reported here as the mean ± the standard deviation. Abbreviations: $R_{in}$ - input resistance, RMP - resting membrane potential, Thre. - spike threshold, Rheo. - rheobase current, fAHP - fast afterhyperpolarization potential, ADP - afterdepolarization potential, $AP_{1/2}$ - action potential half-width, Freq. - maximum recorded spike frequency, $AP_{max}$ - maximum spike height at rheobase, Spkdelay - delay to spike onset.

Supplementary file 2. Select morphological properties of CLA neurons. All values reported here as the mean ± the standard deviation. Abbreviations: #Den. - number of primary dendrites, #Nodes - number of dendritic nodes,<Den.> - average dendritic length, Max Den.$_{order}$ - the highest order dendrite,<Seg.> - average dendritic segment length, ND - node density, Den.$_{max}$ - longest dendrite, Len.$_{max}$ - the polar bin of the longest dendrite.

Supplementary file 3. Response probability of CLA electrophysiological cell types to inputs from individual cortices during optogenetic stimulation. Values here are reported as the number of responsive neurons of a given type divided by the total number that type recorded in all optogenetics experiments involving inputs from a given cortex. Entries with * indicate experiments in which <4 neurons of that cell type were recorded.

MDAR checklist

## Data availability

Processed data and the functions used to generate the figure panels in this study are available at https://github.com/AMShelton/claustrum-integration (copy archived at *Shelton, 2025*).

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
