## [Editor Report · eLife Assessment]

This study compiles a wide range of results on the connectivity, stimulus selectivity, and potential role of the claustrum in sensory behavior. While most of the connectivity results confirm earlier studies, this **valuable** work provides **incomplete** evidence that the claustrum responds to multimodal stimuli and that local connectivity is reduced across cells that have similar long-range connectivity. The conclusions drawn from the behavioral results are weakened by the animals' poor performance on the designed task. This study has the potential to be of interest to neuroscientists.

---

## [Referee Report · Reviewer #1 (Public review)]

Summary:

The paper by Shelton et al investigates some of the anatomical and physiological properties of the mouse claustrum. First, they characterize the intrinsic properties of claustrum excitatory and inhibitory neurons and determine how these different claustrum neurons receive input from different cortical regions. Next, they perform in vitro patch clamp recordings to determine the extent of intraclaustrum connectivity between excitatory neurons. Following these experiments, in vivo axon imaging was performed to determine how claustrum-retrosplenial cortex neurons are modulated by different combinations of auditory, visual, and somatosensory input. Finally, the authors perform claustrum lesions to determine if claustrum neurons are required for performance on a multisensory discrimination task

Strengths:

An important potential contribution the authors provide is the demonstration of intra-claustrum excitation. In addition, this paper does provide the first experimental data where two cortical inputs are independently stimulated in the same experiment (using 2 different opsins). Overall, the in vitro patch clamp experiments and anatomical data provide confirmation that claustrum neurons receive convergent inputs from areas of frontal cortex. These experiments were conducted with rigor and are of high quality.

Weaknesses:

The title of the paper states that claustrum neurons integrate information from different cortical sources. However, the authors did not actually test or measure integration in the manuscript. They do show physiological convergence of inputs on claustrum neurons in the slice work. Testing integration through simultaneous activation of inputs was not performed. The convergence of cortical input has been recently shown by several other papers (Chia et al), and the current paper largely supports these previous conclusions. The in vivo work did test for integration, because simultaneous sensory stimulations were performed. However, integration was not measured at the single cell (axon) level because it was unclear how activity in a single claustrum ROI changes in response to (for example) visual, tactile, and visual-tactile stimulations. Reading the discussion, I also see the authors speculate that the sensory responses in the claustrum could arise from attentional or salience related inputs from an upstream source such as the PFC. In this case, claustrum cells would not integrate anything (but instead respond to PFC inputs).

The different experiments in different figures often do not inform each other. For example, the authors show in Figure 3 that claustrum-RSP cells (CTB cells) do not receive input from the auditory cortex. But then, in Figure 6 auditory stimuli are used. Not surprisingly, claustrum ROIs respond very little to auditory stimuli (the weakest of all sensory modalities). Then, in Figure 7 the authors use auditory stimuli in the multisensory task. It seems that these experiments were done independently and were not used to inform each other.

One novel aspect of the manuscript is the focus on intraclaustrum connectivity between excitatory cells (Figure 2). The authors used wide-field optogenetics to investigate connectivity. However, the use paired patch clamp recordings remains the ground truth technique for determining the rate of connectivity between cell types, and paired recordings were not performed here. It is difficult to understand and gain appreciation for intraclaustrum connectivity when only wide-field optogenetics is used.

In Figure 2, CLA-rsp cells express Chrimson, and the authors removed cells from the analysis with short latency responses (which reflect opsin expression). But wouldn't this also remove cells that express opsin and receive monosynaptic inputs from other opsin expressing cells, therefore underestimating the connectivity between these CLA-rsp neurons? I think this needs to be addressed.

In Figure 5J the lack of difference in the EPSC-IPSC timing in the RSP is likely due to 1 outlier EPSC at 30ms which is most likely reflecting polysynaptic communication. Therefore, I do not feel the argument being made here with differences in physiology is particularly striking.

In the text describing Figure 5, the authors state "These experiments point to a complex interaction ....likely influenced by cell type of CLA projection and intraclaustral modules in which they participate". How does this slice experiment stimulating axons from one input relate to different CLA cell types or intra-claustrum circuits? I don't follow this argument.

In Figure 6G and H the blank condition yields a result similar to many of the sensory stimulus conditions. This blank condition (when no stimulus was presented) serves as a nice reference to compare the rest of the conditions. However, the remainder of the stimulation conditions were not adjusted relative to what would be expected by chance. For example, the response of each cell could be compared to a distribution of shuffled data, where time-series data are shuffled in time by randomly assigned intervals and a surrogate distribution of responses generated. This procedure is repeated 200-1000x to generate a distribution of shuffled responses. Then the original stimulus triggered response (1s post) could be compared to shuffled data. Currently, the authors just compare pre/post mean data using a Mann Whitney test from the mean overall response, which could be biased by a small number of trials. Therefore, I think a more conservative and statistically rigorous approach is warranted here, before making the claim of a 20% response probability or 50% overall response rate.

Regarding Figure 6, a more conventional way to show sensory responses is to display a heatmap of the z-scored responses across all ROIs, sorted by their post-stimulus response. This enables the reader to better visualize and understand the claims being made here, rather than relying on the overall mean which could be influenced by a few highly responsive ROIs.

For Figure 6 it would also help to display some raw data showing responses at the single ROI level and the population level. If these sensory stimulations are modulating claustrum neurons, then this will be observable on the mean population vector (averaged df/f across all ROIs as a function of time) within a given experiment and would add support to the conclusions being made.

As noted by the authors, there is substantial evidence in the literature showing that motor activity arises in mice during these types of sensory stimulation experiments. It is foreseeable that at least some of the responses measured here arise from motor activity. It would be important to identify to what extent this is the case.

All claims in the results for Figure 6 such as "the proportion of responsive axons tended to be highest when stimuli were combined" should be supported by statistics.

For Figure 7, the authors state that mice learned the structure of the task. How is this the case, when the number of misses are 5-6x greater than the number of hits on audiovisual trials (S Fig 19). I don't get the impression that mice perform this task correctly. As shown in Figure 7I, the hit rate is exceptionally low on the audiovisual port in controls. I just can't see how control and lesion mice can have the same hit rate and false alarm rate yet have different d'. Indeed, I might be missing something in the analysis. However, given that both groups of mice are not performing the task as designed, I fail to see how the authors claim regarding multisensory integration by the claustrum is supported. Even if there is some difference in the d' measure, what does that matter when the hits are the least likely trial outcome here for both groups.

In the discussion, it is stated that "While axons responded inconsistently to individual stimulus presentations, their responsivity remained consistent between stimuli and through time on average...". I do not understand this part of the sentence. Does this mean axons are consistently inconsistent?

In the discussion the authors state their axon imaging results contrast with recent studies in mice. Why not actually do the same analysis that Ollerenshaw did, so this statement is supported by fact? As pointed out above, the criteria used to classify an axon as responsive to stimuli was very liberal in this current manuscript.

I find the discussion wildly speculative and broad. For example, "the integrative properties of the CLA could act as a substrate for transforming the information content of its inputs (e.g. reducing trial to trial variability of responses to conjunctive stimuli...)". How would a claustrum neuron responding with a 10% reliability to a stimuli (or set of stimuli) provide any role in reducing trial to trial variability of sensory activity in the cortex?

Comments on the latest version: The authors have revised the manuscript, by adding 1 new supplementary figure, and some minor changes to the text. Overall, my comments regarding the manuscript were not sufficiently addressed. Here is one example:

The authors don't seem to be taking the comments regarding the statistical significance of the sensory responses seriously. If there is a response in 10% of the axons in the blank condition, and a 11 % response in the auditory stimulation, then that means that it is more accurate to say that 1% of axons actually respond to auditory stimulation. "leaving to reader to make their own decisions" as the authors suggest, but then having authors read text such as "All modalities could evoke responses in at least some claustrum neurons", is misleading because no attempt was made to correct for a chance level of detection that is clearly observed in the blank condition. Another interpretation of the authors data would be that in the case of the auditory/visual/somatosensory combined stimuli resulted in 21%(observed) - 10% (blank) = 11% of axons. Therefore, a conclusion that more accurately reflects the data would be that 89% of claustrum axons do not respond, even when the mouse received multisensory stimuli. I tried to get the authors to run some basic stats to more accurately test the true degree of responsiveness, but these changes did not appear in the manuscript.

---

## [Referee Report · Reviewer #2 (Public review)]

Summary:

In this manuscript, Shelton et al. explore the organization of the Claustrum. To do so, they focus on a specific claustrum population, the one projecting to the retrosplenial cortex (CLA-RSP neurons). Using elegant technical approach, they first described electrophysiological properties of claustrum neurons, including the CLA-RSP ones. Further, they showed that CLA-RSP neurons (1) directly excite other CLA neurons, in a 'projection-specific' pattern, i.e. CLA-RSP neurons mainly excite claustrum neurons not projecting to the RSP and (2) received excitatory inputs from multiple cortical territories (mainly frontal ones). In an effort to confirm the 'integrative' property of claustrum networks, they then imaged claustrum axons in the cortex during single- or multi-sensory stimulations. Finally, they investigated the effect of CLA-RSP lesion on performance in a sensory detection task.

Strengths:

Overall, this is a really good study, using state of the art technical approaches to probe the local/global organization of the Claustrum. The in-vitro part is impressive, and the results are compelling.

Weaknesses:

One noteworthy concern arises from the terminology used throughout the study. The authors claimed that the claustrum is an integrative structure. Yet, integration has a specific meaning, i.e. the production of a specific response by a single neuron (or network) in response to a specific combination of several input signals. In this study, the authors showed compelling results in favor of convergence rather than integration. On a lighter note, the in-vivo data are less convincing, and do not entirely support the claim of "integration" made by the authors.

---

## [Referee Report · Reviewer #3 (Public review)]

Public review:

The claustrum is one of the most enigmatic regions of the cerebral cortex, with a potential role in consciousness and integrating multisensory information. Despite extensive connections with almost all cortical areas, its functions and mechanisms are not well understood. In an attempt to unravel these complexities, Shelton et al. employed advanced circuit mapping technologies to examine specific neurons within the claustrum. They focused on how these neurons integrate incoming information and manage the output. Their findings suggest that claustrum neurons selectively communicate based on cortical projection targets and that their responsiveness to cortical inputs varies by cell type.

Imaging studies demonstrated that claustrum axons respond to both single and multiple sensory stimuli. Extended inhibition of the claustrum significantly reduced animals' responsiveness to multisensory stimuli, highlighting its critical role as an integrative hub in the cortex.

However, the study's conclusions at times rely on assumptions that may undermine their validity. For instance, the comparison between RSC projecting and non-RSC projecting neurons is problematic due to potential false negatives in the cell labeling process, which might not capture the entire neuron population projecting to a brain area. This issue casts doubt on the findings related to neuron interconnectivity and projections, suggesting that the results should be interpreted with caution. The study's approach to defining neuron types based on projection could benefit from a more critical evaluation or a broader methodological perspective.

Nevertheless, the study sets the stage for many promising future research directions. Future work could particularly focus on exploring the functional and molecular differences between E1 and E2 neurons and further assess the implications of the distinct responses of excitatory and inhibitory claustrum neurons for internal computations. Additionally, adopting a different behavioral paradigm that more directly tests the integration of sensory information for purposeful behavior could also prove valuable.

---

## [Author Response]

**Author response**

The following is the authors’ response to the original reviews.

**eLife Assessment**
This study compiles a wide range of results on the connectivity, stimulus selectivity, and potential role of the claustrum in sensory behavior. While most of the connectivity results confirm earlier studies, this valuable work provides incomplete evidence that the claustrum responds to multimodal stimuli and that local connectivity is reduced across cells that have similar long-range connectivity. The conclusions drawn from the behavioral results are weakened by the animals' poor performance on the designed task.This study has the potential to be of interest to neuroscientists.

We thank the editor and the reviewers for their feedback on our work, which we have incorporated to help improve interpretation of our findings as outlined in the response below. While we agree with the editor that further work is necessary to provide a comprehensive understanding of claustrum circuitry and activity, this is true of most scientific endeavors and therefore we feel that describing this work as “incomplete” unfairly mischaracterizes the intent of the experiments performed which provide fundamental insights into this poorly understood brain region. Additionally, as identified in the main text, methods section, and our responses to the comments below, we disagree that the behavioral results are “weakened” by the performance of the animals. Our goal was to assess what information animals learned and used in an ambiguous sensory/reward environment, *not* to shape them toward a particular behavior and interpret the results solely based on their accuracy in performing the task.

**Public Reviews:**

**Reviewer #1 (Public Review):**
Summary:The paper by Shelton et al investigates some of the anatomical and physiological properties of the mouse claustrum. First, they characterize the intrinsic properties of claustrum excitatory and inhibitory neurons and determine how these different claustrum neurons receive input from different cortical regions. Next, they perform in vitro patch clamp recordings to determine the extent of intraclaustrum connectivity between excitatory neurons. Following these experiments, in vivo axon imaging was performed to determine how claustrum-retrosplenial cortex neurons are modulated by different combinations of auditory, visual, and somatosensory input. Finally, the authors perform claustrum lesions to determine if claustrum neurons are required for performance on a multisensory discrimination taskStrengths:An important potential contribution the authors provide is the demonstration of intra-claustrum excitation. In addition, this paper provides the first experimental data where two cortical inputs are independently stimulated in the same experiment (using 2 different opsins). Overall, the in vitro patch clamp experiments and anatomical data provide confirmation that claustrum neurons receive convergent inputs from areas of the frontal cortex. These experiments were conducted with rigor and are of high quality.

We thank the reviewer for their positive appraisal of our work.

Weaknesses:The title of the paper states that claustrum neurons integrate information from different cortical sources. However, the authors did not actually test or measure integration in the manuscript. They do show physiological convergence of inputs on claustrum neurons in the slice work. Testing integration through simultaneous activation of inputs was not performed. The convergence of cortical input has been recently shown by several other papers (Chia et al), and the current paper largely supports these previous conclusions. The in vivo work did test for integration because simultaneous sensory stimulations were performed. However, integration was not measured at the single cell (axon) level because it was unclear how activity in a single claustrum ROI changes in response to (for example) visual, tactile, and visual-tactile stimulations. Reading the discussion, I also see the authors speculate that the sensory responses in the claustrum could arise from attentional or salience-related inputs from an upstream source such as the PFC. In this case, claustrum cells would not integrate anything (but instead respond to PFC inputs).

We thank the reviewer for raising this point. In response, we have provided a definition of “integration” in the manuscript text (lines 112-114, 353-354):

“...single-cell responsiveness to more than one input pathway, e.g. being capable of combining and therefore integrating these inputs.”

The reviewer’s point about testing simultaneous input to the claustrum is well made but not possible with the dual-color optogenetic stimulation paradigm used in our study as noted in the Results and Discussion sections (see also Klapoetke et al., 2014, Hooks et al., 2015). The novelty of our paper comes from testing these connections in *single CLA neurons*, something not shown in other studies to-date (Chia et al., 2020; Qadir et al., 2022), which average connectivity over many neurons.

Finally, we disagree with the reviewer regarding whether integration was tested at the single-axon level and provide data and supplementary figures to this effect (**Fig. 6, Supp. Fig. S14**, lines 468-511) . Although the possibility remains that sensory-related information may arise in the prefrontal cortex, as we note, there is still a large collection of studies (including this one) that document and describe direct sensory inputs to the claustrum (Olson & Greybeil, 1980; Sherk & LeVay, 1981; Smith & Alloway, 2010; Goll et al., 2015; Atlan et al., 2017; etc.). We have updated the wording of these sections to note that both direct and indirect sensory input integration is possible.

The different experiments in different figures often do not inform each other. For example, the authors show in Figure 3 that claustrum-RSP cells (CTB cells) do not receive input from the auditory cortex. But then, in Figure 6 auditory stimuli are used. Not surprisingly, claustrum ROIs respond very little to auditory stimuli (the weakest of all sensory modalities). Then, in Figure 7 the authors use auditory stimuli in the multisensory task. It seems that these experiments were done independently and were not used to inform each other.

The intention behind the current manuscript was to provide a deep characterisation of claustrum to inform future research into this enigmatic structure. In this case, we sought to test pathways in vivo that were identified as being weak or absent in vitro to confirm and specifically rule out their influence on computations performed by claustrum. We agree with the reviewer’s assessment that it is not surprising that claustrum ROIs respond weakly to auditory stimuli. Not testing these connections in vivo because of their apparent sparsity in vitro would have represented a critical gap in our knowledge of claustrum responses during passive sensory stimulation.

One novel aspect of the manuscript is the focus on intraclaustrum connectivity between excitatory cells (Figure 2). The authors used wide-field optogenetics to investigate connectivity. However, the use of paired patch-clamp recordings remains the ground truth technique for determining the rate of connectivity between cell types, and paired recordings were not performed here. It is difficult to understand and gain appreciation for intraclaustrum connectivity when only wide-field optogenetics is used.

We thank the reviewer for acknowledging the novelty of these experiments. We further acknowledge that paired patch-clamp recordings are the gold standard for assessing synaptic connectivity. Typically such experiments are performed in vitro, a necessity given the ventral location of claustrum precluding in vivo patching. In vitro slice preparations by their very nature sever connections and lead to an underestimate of connectivity as noted in our Discussion. Kim et al. (2016) have done this experiment in coronal slices with the understanding that excitatory-excitatory connectivity would be local (<200 μm) and therefore preserved. We used a variety of approaches that enabled us to explore connectivity along the longitudinal axis of the brain (the rostro-caudal, e.g. “long” axis of the claustrum), providing fresh insight into the circuitry embedded within this structure that would be challenging to examine using dual recordings. Further, our optogenetic method (CRACM, Petreanu et al., 2007), has been used successfully across a variety of brain structures to examine excitatory connectivity while circumventing artifacts arising from the slice axis.

In Figure 2, CLA-rsp cells express Chrimson, and the authors removed cells from the analysis with short latency responses (which reflect opsin expression). But wouldn't this also remove cells that express opsin and receive monosynaptic inputs from other opsin-expressing cells, therefore underestimating the connectivity between these CLA-rsp neurons? I think this needs to be addressed.

The total number of opsin-expressing CLA neurons in our dataset is 4/46 tested neurons. Assuming all of these neurons project to RSP, they would have accounted for 4/32 CLARSP neurons. Given the rate of monosynaptic connectivity observed in this study, these neurons would only contribute 2-3 additional connected neurons. Therefore, the exclusion of these neurons does not significantly impact the overall statistical accuracy of our connectivity findings.

In Figure 5J the lack of difference in the EPSC-IPSC timing in the RSP is likely due to 1 outlier EPSC at 30 ms which is most likely reflecting polysynaptic communication. Therefore, I do not feel the argument being made here with differences in physiology is particularly striking.

We thank the reviewer for their attention to detail about this analysis. We have performed additional statistics and found that leaving this neuron out does not affect the significance of the results (new p-value = 0.158, original p-value = 0.314, Mann-Whitney U test). We have removed this datapoint from the figure and our analysis.

In the text describing Figure 5, the authors state "These experiments point to a complex interaction ....likely influenced by cell type of CLA projection and intraclaustral modules in which they participate". How does this slice experiment stimulating axons from one input relate to different CLA cell types or intra-claustrum circuits? I don't follow this argument.

We have removed this speculation from the Results section.

In Figure 6G and H, the blank condition yields a result similar to many of the sensory stimulus conditions. This blank condition (when no stimulus was presented) serves as a nice reference to compare the rest of the conditions. However, the remainder of the stimulation conditions were not adjusted relative to what would be expected by chance. For example, the response of each cell could be compared to a distribution of shuffled data, where time-series data are shuffled in time by randomly assigned intervals and a surrogate distribution of responses generated. This procedure is repeated 200-1000x to generate a distribution of shuffled responses. Then the original stimulus-triggered response (1s post) could be compared to shuffled data. Currently, the authors just compare pre/post-mean data using a Mann-Whitney test from the mean overall response, which could be biased by a small number of trials. Therefore, I think a more conservative and statistically rigorous approach is warranted here, before making the claim of a 20% response probability or 50% overall response rate.

We appreciate the reviewer's thorough analysis and suggestion for a more conservative statistical approach. We acknowledge that responses on blank trials occur about 10% of the time, indicating that response probabilities around this level may not represent "real" responses. To address this, we will include the responses to the blank condition in the manuscript (lines 505-509). This will allow readers to make informed decisions based on the presented data.

Regarding Figure 6, a more conventional way to show sensory responses is to display a heatmap of the z-scored responses across all ROIs, sorted by their post-stimulus response. This enables the reader to better visualize and understand the claims being made here, rather than relying on the overall mean which could be influenced by a few highly responsive ROIs.

We apologize to the reviewer that our data in this figure was challenging to interpret. We have included an additional supplemental figure (Supp. Fig. S15) that displays the requested information.

For Figure 6, it would also help to display some raw data showing responses at the single ROI level and the population level. If these sensory stimulations are modulating claustrum neurons, then this will be observable on the mean population vector (averaged df/f across all ROIs as a function of time) within a given experiment and would add support to the conclusions being made.

We appreciate the reviewer’s desire to see more raw data – we would have included this in the figure given more space. However, the average df/f across all ROIs is shown as a time series with 95% confidence intervals in Fig. 6D.

As noted by the authors, there is substantial evidence in the literature showing that motor activity arises in mice during these types of sensory stimulation experiments. It is foreseeable that at least some of the responses measured here arise from motor activity. It would be important to identify to what extent this is the case.

While we acknowledge that some responses may arise from motor-related activity, addressing this comprehensively is beyond the scope of this paper. Given the extensive number of trials and recorded axonal segments, we believe that motor-related activity is unlikely to significantly impact the average response across all trials. Future studies focusing specifically on motor activity during sensory stimulation experiments would be needed to elucidate this aspect in detail.

All claims in the results for Figure 6 such as "the proportion of responsive axons tended to be highest when stimuli were combined" should be supported by statistics.

We have provided additional statistics in this section (lines 490-511) to address the reviewer’s comment.

In Figure 7, the authors state that mice learned the structure of the task. How is this the case, when the number of misses is 5-6x greater than the number of hits on audiovisual trials (S Figure 19). I don't get the impression that mice perform this task correctly. As shown in Figure 7I, the hit rate is exceptionally low on the audiovisual port in controls. I just can't see how control and lesion mice can have the same hit rate and false alarm rate yet have different d'. Indeed, I might be missing something in the analysis. However, given that both groups of mice are not performing the task as designed, I fail to see how the authors' claim regarding multisensory integration by the claustrum is supported. Even if there is some difference in the d' measure, what does that matter when the hits are the least likely trial outcome here for both groups.

We thank the reviewer for their comments and hope the following addresses their confusion about the performance of animals during our multimodal conditioning task.

Firstly, as pointed out by the reviewer, the hit-rate (HR) is lower than false-alarm-rate (FR) but crucially only when assessed explicitly within-condition (e.g. just auditory or just visual stimulation). Given the multimodal nature of the assay, HR and FR could also be evaluated across different trials, unimodal and multimodal, for both auditory and visual stimuli. Doing so resulted in a net positive d', as observed by the reviewer. From this perspective, and as documented in the Methods (Multimodal Conditioning and Reversal Learning) and Supplemental Figures, mice do indeed learn the conditioning task and perform at above-chance levels.

Secondly, as raised in the Discussion, an important caveat of this assay was that it was unnecessary for mice to learn the task structure explicitly but, rather, that they respond to environmental cues in a reward-seeking manner that indicated perception of a stimulus. "Performance" as it is quantified here demonstrates a perceptual difference between conditions that is observed through behavioral choice and timing, not necessarily the degree to which the mice have an understanding of the task per se.

In the discussion, it is stated that "While axons responded inconsistently to individual stimulus presentations, their responsivity remained consistent between stimuli and through time on average...". I do not understand this part of the sentence. Does this mean axons are consistently inconsistent?

The reviewer’s interpretation is correct – although recorded axons tended to have a preferred stimulus or combination of stimuli, they displayed variability in their responses (response probability), though little or no variability in their likelihood to respond over time (on average).

In the discussion, the authors state their axon imaging results contrast with recent studies in mice. Why not actually do the same analysis that Ollerenshaw did, so this statement is supported by fact? As pointed out above, the criteria used to classify an axon as responsive to stimuli were very liberal in this current manuscript.

While we appreciate this comment from the reviewer, we feel that it was not necessary to perform similar analyses to those of Ollerenshaw et al in order to appreciate that methodological differences between these studies would have confounded any comparisons made, as we note in the Discussion.

I find the discussion wildly speculative and broad. For example, "the integrative properties of the CLA could act as a substrate for transforming the information content of its inputs (e.g. reducing trial-to-trial variability of responses to conjunctive stimuli...)". How would a claustrum neuron responding with a 10% reliability to a stimuli (or set of stimuli) provide any role in reducing trial-to-trial variability of sensory activity in the cortex?

We thank the reviewer for their feedback. We acknowledge the reviewer's concern regarding the speculative nature of our discussion. To address the specific point raised, while a neuron with a 10% reliability might appear limited in reducing trial-to-trial variability in sensory activity, it's possible that such neurons are responsive to a combination of stimuli or conditions not fully controlled or recorded in our current setup. For instance, variables like the animal’s attentional or motivational states could influence the responsiveness of claustrum neurons, thus integrating these inputs could theoretically modulate cortical processing. We have refined this section to clarify these points (now lines 810-813).

**Reviewer #2 (Public Review):**
Summary:In this manuscript, Shelton et al. explore the organization of the Claustrum. To do so, they focus on a specific claustrum population, the one projecting to the retrosplenial cortex (CLA-RSP neurons). Using an elegant technical approach, they first described electrophysiological properties of claustrum neurons, including the CLA-RSP ones. Further, they showed that CLA-RSP neurons (1) directly excite other CLA neurons, in a 'projection-specific' pattern, i.e. CLA-RSP neurons mainly excite claustrum neurons not projecting to the RSP and (2) receive excitatory inputs from multiple cortical territories (mainly frontal ones). To confirm the 'integrative' property of claustrum networks, they then imaged claustrum axons in the cortex during singleor multi-sensory stimulations. Finally, they investigated the effect of CLA-RSP lesion on performance in a sensory detection task.Strengths:Overall, this is a really good study, using state-of-the-art technical approaches to probe the local/global organization of the Claustrum. The in-vitro part is impressive, and the results are compelling.

We thank the reviewer for their positive appraisal of our work.

Weaknesses:One noteworthy concern arises from the terminology used throughout the study. The authors claimed that the claustrum is an integrative structure. Yet, integration has a specific meaning, i.e. the production of a specific response by a single neuron (or network) in response to a specific combination of several input signals. In this study, the authors showed compelling results in favor of convergence rather than integration. On a lighter note, the in-vivo data are less convincing, and do not entirely support the claim of "integration" made by the authors.

We thank the reviewer for their clarity on this issue. We absolutely agree that without clear definition in the study, interpretation of our data could be misconstrued for one of several possible meanings. We have updated our Introduction, Results, and Discussion text to reflect the definition of ‘integration’ we used in the interpretation of our work and hope this clarifies our intent to the reader.

**Reviewer #3 (Public Review):**
The claustrum is one of the most enigmatic regions of the cerebral cortex, with a potential role in consciousness and integrating multisensory information. Despite extensive connections with almost all cortical areas, its functions and mechanisms are not well understood. In an attempt to unravel these complexities, Shelton et al. employed advanced circuit mapping technologies to examine specific neurons within the claustrum. They focused on how these neurons integrate incoming information and manage the output. Their findings suggest that claustrum neurons selectively communicate based on cortical projection targets and that their responsiveness to cortical inputs varies by cell type.Imaging studies demonstrated that claustrum axons respond to both single and multiple sensory stimuli. Extended inhibition of the claustrum significantly reduced animals' responsiveness to multisensory stimuli, highlighting its critical role as an integrative hub in the cortex.However, the study's conclusions at times rely on assumptions that may undermine their validity. For instance, the comparison between RSC-projecting and non-RSC-projecting neurons is problematic due to potential false negatives in the cell labeling process, which might not capture the entire neuron population projecting to a brain area. This issue casts doubt on the findings related to neuron interconnectivity and projections, suggesting that the results should be interpreted with caution. The study's approach to defining neuron types based on projection could benefit from a more critical evaluation or a broader methodological perspective.

We thank the reviewer for their attention to the methods used in our study. We acknowledge that there is an inherent bias introduced by false-negatives as a result of incomplete labeling but contend that this is true of most modern tracing experiments in neuroscience, irrespective of the method used. Moreover, if false-negative biases are affecting our results, then they likely do so in the direction of supporting our findings – perfect knowledge of claustrum connectivity would likely enhance the effects seen by increasing the pool of neurons for which we find an effect. For example, our cortico-claustal connectivity findings in Figure 3 likely would have shown even larger effects should false-negative CLARSP neurons have been positively identified.

Where appropriate we have provided estimates of variability and certainty in our experimental findings and do not claim any definitive knowledge of the true rate and scope of claustrum connectivity.

Nevertheless, the study sets the stage for many promising future research directions. Future work could particularly focus on exploring the functional and molecular differences between E1 and E2 neurons and further assess the implications of the distinct responses of excitatory and inhibitory claustrum neurons for internal computations. Additionally, adopting a different behavioral paradigm that more directly tests the integration of sensory information for purposeful behavior could also prove valuable.

We thank the reviewer for their outlook on the future directions of our work. These avenues for study, we believe, would be very fruitful in uncovering the cell-type-specific computations performed by claustrum neurons.

**Recommendations for the authors:**

**Reviewing Editor (Recommendations for the Authors):**
The editor recommends addressing the issues raised by the reviewers about the statistical significance of sensory response with respect to blank stimuli, and solving the issue generated by the exclusion of monosynaptically connected neurons in the connectivity study, to raise the assessment strength of evidence from incomplete to solid. Moreover, as the reported result stands, the behavioral task does not seem to be learned by the animals as the animals are above chance for visual and auditory but largely below chance level for multisensory. It seems that the animals do not perform a multisensory task. The authors should clarify this.
**Reviewer #1 (Recommendations For The Authors):**
Several references were missing from the manuscript, where mouse CLA-retrosplenial or CLA-frontal neurons were investigated and would be highly relevant to both the discussion of claustrum function and the context of the methodologies used here. (Wang et al., 2023 Nat Comm; Nair et al., 2023 PNAS, Marriott et al. 2024 Cell Reports ; Faig et al., 2024 Current Biology).
**Reviewer #2 (Recommendations For The Authors):**
Let me be clear, this is an excellent study, using state-of-the-art technical approaches to probe the local/global organization of the Claustrum. However, the study is somehow disconnected, with a fantastic in-vitro part, and, in my opinion, a less convincing in-vivo one.As stated in the public review, I'm concerned about the use of the term "integration", as, in my opinion, the data presented in this study (which I repeat are of excellent level) do not support that claim.Below are my main points regarding the article:(1) My main comment relates to the use of the term 'integration'. It might be a semantic debate, but I think that this is an important one. In my opinion, neural integration is the "summing of several neural input signals by a single neuron to produce an output signal that is some function of those inputs". As the authors state in the discussion, they were not able to "assess the EPSP response magnitude to the conjunction of stimuli due to photosensitivity of ChrimsonR opsins to blue light". Therefore, the authors did not specifically prove integration, but rather input convergence. This does not mean that the results presented are not important or of excellent quality, but I encourage the authors to either tone down the part on integration or to give a clear definition of what they call integration.(2) The in vivo imaging data are somehow confusing. First, the authors image two claustral populations simultaneously (the CLA-RSP and the CLA-ACA axons). I may be missing the information, but there is no evidence that these cells overlap in the CLA (no data in the supplement and existing literature only support partial overlap). Second, in the results part, the authors claim that 96% of the sensory-responsive axons displayed multisensory response. This, combined with the 47% of axons responsive to at least one stimulus should lead to a global response of around 45% of the axons in multisensory trials. Yet, in Figures 6F-G, one can see that the response probability is actually low (closer to 20%). To be honest, I cannot really understand how to make sense of these results. At first, I thought that most of the multisensory responsive axons show no response during multisensory stimulus (but one in the unimodal stimulus). This hypothesis is however unlikely, as response AUC is biased toward positivity in Figure 6H. Overall, I'm not totally convinced by the imaging data, and I think that the authors should be more cautious about interpreting their results (as they are in the discussion part, but less in the results part).(3) The TetTox approach used in the study ablates all neurons expressing the CRE in the CLA. If the hypothesis proposed by the authors is true, then ablating one subpopulation should not impact that much the functioning of the whole CLA, as other neurons will likely "integrate" information coming from multiple cortices (Figures 3 and 4), the local divergence (Figure 1) will then allow the broadcasting of this information back to multiples cortices. Do the authors think that such an approach deeply modified intra-claustral network connectivity? If this is not the case, shouldn't we expect less effect after lesioning a specific sub-population of CLA neurons?(4) The behavioral protocol is also confusing. If I understand correctly, the aim of the task was to probe the D-Prime factor, as all trials, whatever the response of the animal are rewarded. From the Figure 7I, one can see that the mice cannot properly answer to the audiovisual cues, clearly indicating that both groups show impaired response to this type of trial. The whole conclusion of the authors is therefore drawn from the D-Prime calculation. However, even if D-Prime should represent a measure of sensitivity (i.e. is unaffected by response bias), two assumptions need to be met: (1) the signal and noise distributions should be both normal, and (2) the signal and noise distributions should have the same standard deviation. However, these assumptions cannot be tested in the task used by the authors (one would need rating tasks). The authors might want to use nonparametric measures of sensitivity such as A' (see Pollack and Norman 1964).
**Reviewer #3 (Recommendations For The Authors):**
While the study is comprehensive, some of its conclusions are based on assumptions that potentially weaken their validity. A significant issue arises in the comparison between neurons that project to the retrosplenial cortex (RSC) and those that do not. This differentiation is based on retrograde labeling from a single part of the RSC. However, CTB labeling, the technique used, does not capture 100% of the neurons projecting to a brain area. The study itself demonstrates this by showing that injecting the dye into three sections of the RSC results in three overlapping populations of neurons in the claustrum. Therefore, limiting the injection to just one of these areas inevitably leads to many false negatives-neurons that project to the RSC but are not marked by the CTB. This issue recurs in the analysis of neurons projecting to both the RSC and the prelimbic cortex (PL), where assumptions about interconnectivity are made without a thorough examination of overlap between these populations. The incomplete labeling complicates the interpretation of the data and draws firm conclusions from it.Minor.There is a reference to Figure 1D where claustrum->cortical connections are described. This should be 5D.

This is a correct reference pointing back to our single-cell characterizations of CLA morphoelectric types.

End of Page 22. Implies should be imply.

This has been resolved in the manuscript text.